# TIME-FFM: Towards LM-Empowered Federated Foundation Model for Time Series Forecasting

**Qingxiang Liu**[1,2]    **Xu Liu**[3]    **Chenghao Liu**[4]    **Qingsong Wen**[5]    **Yuxuan Liang**[1*]

[1] The Hong Kong University of Science and Technology (Guangzhou)
[2] Institute of Computing Technology Chinese Academy of Sciences
[3] National University of Singapore  [4] Salesforce AI Research  [5] Squirrel AI
qingxiangliu737@gmail.com, liuxu@comp.nus.edu.sg
chenghao.liu@salesforce.com, qingsongedu@gmail.com, yuxliang@outlook.com

## Abstract

Unlike natural language processing and computer vision, the development of Foundation Models (FMs) for time series forecasting is blocked due to data scarcity. While recent efforts are focused on building such FMs by unlocking the potential of language models (LMs) for time series analysis, dedicated parameters for various downstream forecasting tasks need training, which hinders the common knowledge sharing across domains. Moreover, data owners may hesitate to share the access to local data due to privacy concerns and copyright protection, which makes it impossible to simply construct a FM on cross-domain training instances. To address these issues, we propose TIME-FFM, a Federated Foundation Model for TIME series forecasting by leveraging pretrained LMs. Specifically, we begin by transforming time series into the modality of text tokens. To bootstrap LMs for time series reasoning, we propose a prompt adaption module to determine domain-customized prompts dynamically instead of artificially. Given the data heterogeneity across domains, we design a personalized federated training strategy by learning global encoders and local prediction heads. Our comprehensive experiments indicate that TIME-FFM outperforms state-of-the-arts and promises effective few-shot and zero-shot forecaster. The code is available at https://github.com/CityMind-Lab/NeurIPS24-Time-FFM/tree/main.

## 1 Introduction

Time series forecasting plays an important role in many real-world application domains [1], such as energy consumption prediction, weather forecasting, and disease transmission. Recently, a multitude of deep learning models have been designed for time series forecasting based on Convolutional Neural Networks [2, 3, 4], Recurrent Neural Networks [5, 6], and Transformers [7, 8, 9, 10]. Inspired by the prominent performance gained by Foundation Models (FMs) in the realms of Natural Language Processing (NLP) [11, 12, 13, 14] and Computer Vision (CV) [15, 16], great research interests have been triggered to build pretrained FMs for time series community [17, 18, 19, 20, 21]. Nonetheless, due to significant time series data scarcity, these FMs are of poor capability to cultivate general representations, failing to promise remarkable fine-tuning or zero-shot performance for diverse downstream forecasting tasks [22, 23]. As a result, a collection of methods have been proposed to borrow the pretrained language FMs to time series community by *cross-modality adaption* [24, 22, 23], thus unlocking the tapped potential of language models (LMs) for time series modeling.

---

*Y. Liang is the corresponding author. This work was done when Q. Liu was an intern at HKUST(GZ).

38th Conference on Neural Information Processing Systems (NeurIPS 2024).

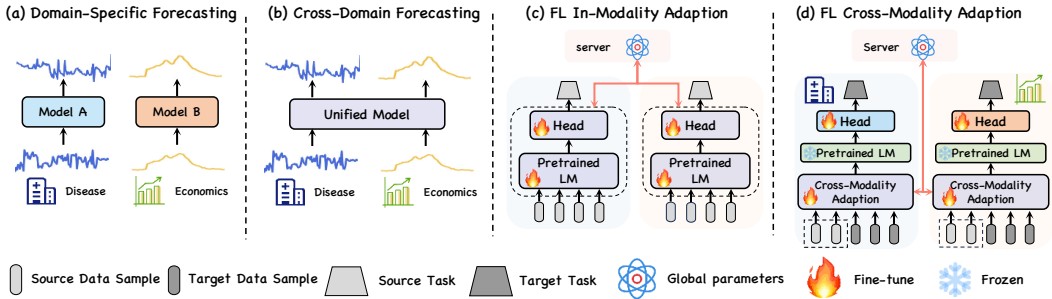

Figure 1: (a) Specific prediction models are trained for diverse domains. (b) A unified model is trained for cross-domain time series. (c) The current in-modality adaption in FL setting fine-tunes LM for NLP tasks, with all the trained parameters are exchanged between clients and the server. (d) Our proposal investigates how to construct a FM by unlocking the potential of LM for cross-domain time series forecasting in FL paradigm.

While these endeavors provide FMs for time series forecasting, the incorporated cross-modality adaption modules and unfrozen components of pretrained LMs need training from scratch for specific domains, thus restricting the mining of underlying temporal commonality in cross-domain time series data. As is shown in Figure 1(a), disease and economics datasets are employed for training the FM respectively to obtain domain-optimal model parameters, hardly generalizing to other domains. [25] proposes to train a unified model (named UniTime) on the mixture of cross-domain time series data (Figure 1(b)), which ensures the cultivation of general-purpose representations, thus promising the zero-shot performance on unseen domains. Despite its effectiveness, they adopt the **centralized** training mode, where the historical records of time series across diverse domains are uploaded to a central server for optimizing the unified model. *Due to copyright protection and privacy concerns, data owners may hesitate to share the access to these domain-specific raw records.*

Federated Learning (FL) [26, 27] provides the mainstream solution for the aforementioned problem, where data owners train prediction models locally and exchange the intermediate model parameters or gradients with the central server, without the disclosure of raw data records. Moreover, in UniTime, a retractable prediction head is introduced to accommodate the heterogeneous output needs whereas FL paradigm makes it possible to construct domain-customized heads. However, current efforts are merely focused on how to fine-tune LMs in federated setting for NLP tasks (i.e., *in-modality adaption of LMs for target tasks in Figure 1(c))* [28, 29, 30, 31], rather than cross-modality adaption of LMs for time series forecasting. The realization of this federated FM is non-trivial technically, given the ubiquitous heterogeneity in cross-domain time series data. **(1) Heterogeneous inputs**: Cross-domain time series data input into the FM are heterogeneous in terms of dimensions and historical readings, posing evident difficulty to modality alignment. **(2) Rigid instructions as prompts**: Prompts are adopted to bootstrap LMs for time series reasoning hinging on rigid domain-specific instructions [25, 22], rather than the understanding of LMs, exhibiting poor robustness for unseen domains. **(3) Conflicts between generalization and personalization**: The ideal FM needs to learn the common temporal representations across domains and simultaneously enable the personalized prediction for domain-specific inputs.

To address the challenges, we propose TIME-FFM, a Federated Foundation Model for TIME series forecasting by repurposing LMs (Figure 1(d)). First, we perform modality alignment by transforming time series data into text tokens to empower the pretrained LM for time series reasoning. Second, we design a prompt adaption module to dynamically determine domain-specific prompts, which can bootstrap the LM for cross-domain time series analysis from the perspective of LM itself, rather than from human cognition by employing hand-crafted instructions as prompts. To tackle the data heterogeneity across domains, we introduce a personalized federated training strategy by learning a global encoder and personalized prediction heads, given the shared representations across domains. Our main contributions are summarized as follows.

- We present the first attempt to build a federated FM for time series forecasting by exploiting the sequence reasoning potential of LMs, avoiding the disclosure of local data.
- We propose TIME-FFM, which firstly aligns the modality from time series data to natural language and adaptively determines prompts to guide the LM for time series reasoning. Moreover, we intro-

duce a personalized FL strategy to strike a balance between sharing common temporal knowledge and ensuring customized prediction results.

- The extensive evaluation results demonstrate that TIME-FFM leads to state-of-the-art performance in mainstream forecasting tasks, especially in few-shot or zero-shot forecasting settings.

## 2 Related Work

**FMs for Time Series Forecasting.** Recent studies have demonstrated the effectiveness of fine-tuning pretrained FMs for various downstream tasks, such as BERT [11], GPT [12], GPT2 [13], and LLaMA [14] in NLP and DEiT [15] and BEiT [16] in CV. Inspired by the success, some efforts have been focused on developing FMs for time series community, such as [17, 18, 21, 32]. However, due to data deficiencies, these pretrained models cannot guarantee the learning of general-purpose representations for time series analysis and hence fail to apply to a multitude of downstream tasks. Another line of researches attempt to leverage pretrained FMs in NLP or CV for time series analysis by cross-modality adaption strategies [33, 34, 35, 24, 23, 22], such as fine-tuning and model reprogramming, which hinges on the powerful generalization capability of Transformers for sequence tokens. [23] freezes the self-attention modules and feedforward layers of GPT2, and only fine-tunes the positional embedding and normalization layers. The proposed GPT4TS outperforms the relevant models in most time series tasks. On the contrary, [22] freezes the LM as a whole and transforms the modality of time series to natural language by patch reprogramming. These methods enable unified model structure rather than unified parameters for diverse downstream tasks, which makes the proposed FMs learn impaired temporal commonality. [25] proposes to train a unified prediction model for cross-domain time series forecasting, which enables to learn the intrinsic temporal patterns. However, the centralized training mode brings privacy concerns for cross-domain data owners and FL paradigm may provide a promising solution.

**Federated Fine-tuning of LMs.** Given the exceptional performance of LMs and the emerging privacy preserving resolutions, incorporating LMs with FL is becoming a popular research trend. There have been some implementation frameworks [36, 29, 37, 38, 39, 40] to support fine-tuning LMs in FL setting. Moreover, considering the immense communication cost, some communication-efficient federated fine-tuning methods have been proposed, such as [38, 41, 30, 28]. A few researches aim to investigate the effects of data heterogeneity on fine-tuning performance, and then propose the personalized federated instruction tuning methods, e.g., [42, 29, 31]. Nonetheless, these methods concentrate on fine-tuning or fully-tuning pretrained LMs in FL paradigm for NLP tasks, but fail to cover the cross-modality adaption of LMs for time series forecasting.

## 3 Methodology

### 3.1 Problem Definition

Given $N$ domains, let $\boldsymbol{x}_{i,t} = \{x_{i,t}^1, \cdots, x_{i,t}^{c_i}\} \in \mathbb{R}^{c_i}$ denote the observation of domain $i$ at the time step $t$, where $c_i$ represents the number of dimensions (channels). In the context of time series forecasting, we denote $\boldsymbol{X}_i = \{\boldsymbol{x}_{i,1}, \cdots, \boldsymbol{x}_{i,L_i}\} \in \mathbb{R}^{L_i \times c_i}$ as the input of the prediction model $f_i(\cdot)$, where $L_i$ represents the domain-variant lookback window. The ground truths can be denoted as $\boldsymbol{Y}_i = \{\boldsymbol{x}_{i,L_i+1}, \cdots, \boldsymbol{x}_{i,L_i+F_i}\} \in \mathbb{R}^{F_i \times c_i}$, where $F_i$ represents the future prediction window. Let $\mathcal{D}_i = \{(\boldsymbol{X}_i; \boldsymbol{Y}_i)\}$ denote the local data set of $i$ and $D_i = |\mathcal{D}_i|$ the data size. Given the set of personalized model parameters $\{w_i\}$, the objective of federated FM for cross-domain time series forecasting can be formulated as

$$\min_{\{w_1, \cdots, w_N\}} \mathcal{L} = \frac{1}{N} \sum_{i=1}^{N} \frac{1}{D_i} \sum_{(\boldsymbol{X}_i; \boldsymbol{Y}_i) \in \mathcal{D}_i} \| \boldsymbol{Y}_i - f_i(w_i; \boldsymbol{X}_i) \|_2^2 . \tag{1}$$

### 3.2 Model Structure

The model architecture is elaborated in Figure 2. Our model encompasses three components: (1) modality alignment and prompt adaption, (2) LM backbone, and (3) prediction head. The modules of modality alignment and prompt adaption are designed for cross-modality alignment and adaptive

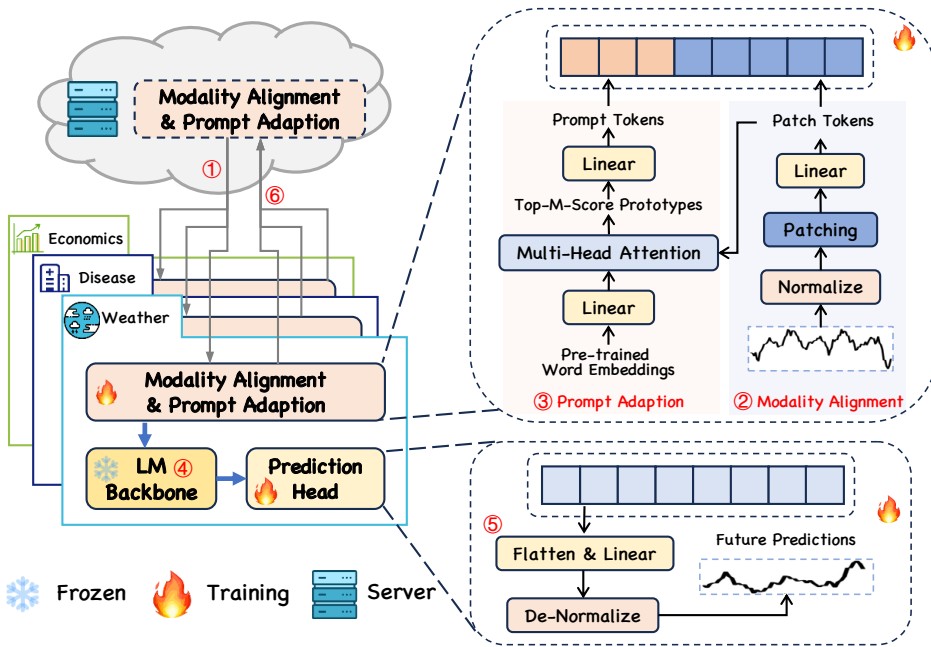

Figure 2: Overall architecture of TIME-FFM. Each round begins with ① downloading global parameters of modality alignment and prompt adaption modules. We ② conduct modality alignment to generate patch tokens and ③ adaptively determine prompt tokens. ④ The two tokens are input into the LM backbone and ⑤ the outputs are projected to generate the prediction results. After local optimization, ⑥ the updated parameters of modality alignment and prompt adaption modules are uploaded to the server for aggregation.

prompt determination. We employ the backbone of GPT2 [13] with freezing all parameters. The prediction head enables domain-specific prediction results.

**Modality Alignment.** Here we transform time series into the modality of text tokens. To accommodate domain-variant channels $c_i$, we adopt the channel-independent strategy [43] to split multivariate time series $\boldsymbol{X}_i$ into $c_i$ univariate series and individually process each. Let $\boldsymbol{X}_i^j = \{x_{i,1}^j, \cdots x_{i,L_i}^j\} \in \mathbb{R}^{1 \times L_i}$ denote the $j$-th univariate series from $\boldsymbol{X}_i$. Then we normalize each series $\boldsymbol{X}_i^j$ to mitigate the effect of distribution diversity [44]. Since each data point of $\boldsymbol{X}_i^j$ does not have explicit semantic knowledge like words in sentences, we adopt the patching technique [43] to segment $\boldsymbol{X}_i^j$ into subseries (termed *patches*), each of which can *aggregate the local information and better retain the temporal knowledge*. Specifically, let $P$ denote the patch length and $S_i$ denote the stride length of domain $i$. Hence, the number of patches can be defined as $B_i = \left\lceil \frac{L_i - P}{S_i} \right\rceil + 1$. We denote $\boldsymbol{X}_{i,S}^j \in \mathbb{R}^{B_i \times P}$ as the generated patches from $\boldsymbol{X}_i^j$. We subsequently employ a linear layer to project the patches into tokens $\hat{\boldsymbol{X}}_{i,S}^j \in \mathbb{R}^{B_i \times D}$, where $D$ is the input dimension size of the LM backbone. $\hat{\boldsymbol{X}}_{i,S}^j$ together with prompt tokens (in the next part) will be input into the LM backbone.

**Prompt Adaption.** In the time series forecasting FMs based on LMs, domain instructions are designed as prompts to complement the patch tokens and inform the LM backbone of domain-specific knowledge [22, 25]. These manually-designed prompts depend completely on experts' knowledge and may vary from each other due to different understandings. Furthermore, according to the results, more detailed instructions can always yield better prediction performance [22], which may make us naturally draw a conclusion that the ultimate performance hinges on the length of prompts. However, longer prompt tokens will present substantial challenge on the computation burden. Different from images [45] or acoustic data [33], which can be "translated" into natural language seamlessly, the manually-crafted prompts are error-prone to describe the characteristics of the raw time series. **To this end, a better way is to design prompts from LM's understandings of the patch tokens rather than human cognition of raw time series data.** Here, we propose to adaptively determine prompts

based on patch tokens from the source corpus of pretrained LM (which includes $V$ pretrained word embeddings, denoted as $\boldsymbol{E} \in \mathbb{R}^{V \times D}$). Similar to [22], we project $\boldsymbol{E}$ to a smaller collection of *text prototypes*, denoted as $\boldsymbol{E}' \in \mathbb{R}^{V' \times D}$ by a linear layer, with $V' \ll V$, to avoid the potential large parameter space. We adopt a modified multi-head attention layer to obtain the correlation between $\boldsymbol{E}'$ and $\hat{\boldsymbol{X}}_{i,S}^j$, and subsequently select $M$ mostly related text prototypes as prompts. Concretely, for each head $h \in \{1, \cdots, H\}$, we have the query matrix $\boldsymbol{Q}_{i,h}^j = \boldsymbol{E}' \boldsymbol{W}_h^Q$ and the key matrix $\boldsymbol{K}_{i,h}^j = \hat{\boldsymbol{X}}_{i,S}^j \boldsymbol{W}_h^K$, where $\boldsymbol{W}_h^Q, \boldsymbol{W}_h^K \in \mathbb{R}^{D \times d}$ and $d = \lfloor \frac{D}{H} \rfloor$. Since we do not aim to return a weighted value matrix according to the given query but merely evaluate the correlation of text prototypes and patch tokens, we omit the value matrix here. The attention score matrix is denoted as $\boldsymbol{O}_{i,h}^j \in \mathbb{R}^{V' \times B_i}$ and can be calculated as

$$\boldsymbol{O}_{i,h}^j = \text{SOFTMAX}\left(\frac{\boldsymbol{Q}_{i,h}^j \boldsymbol{K}_{i,h}^{j\top}}{\sqrt{d}}\right). \tag{2}$$

We obtain $\hat{\boldsymbol{O}}_{i,h}^j \in \mathbb{R}^{V' \times 1}$ by calculating the summation of $\boldsymbol{O}_{i,h}^j$ per row. Each value in $\hat{\boldsymbol{O}}_{i,h}^j$ represents the correlation degree of the corresponding text prototype in $\boldsymbol{E}'$ to all patch tokens $\hat{\boldsymbol{X}}_{i,S}^j$. We select $M$ prototypes from $\boldsymbol{Q}_{i,h}^j$ with top attention scores to form the potential prompts $\boldsymbol{Z}_{i,h}^j \in \mathbb{R}^{M \times d}$, i.e., $\boldsymbol{Z}_{i,h}^j = \boldsymbol{Q}_{i,h}^j \left[ \text{TOPM}(\hat{\boldsymbol{O}}_{i,h}^j) \right]$. We can obtain $\boldsymbol{Z}_i^j \in \mathbb{R}^{M \times D}$ by aggregating $\boldsymbol{Z}_{i,h}^j$ from all $H$ heads. Finally, we employ a linear layer to project $\boldsymbol{Z}_i^j$ to the prompt tokens $\hat{\boldsymbol{Z}}_i^j \in \mathbb{R}^{M \times D}$.

**Prediction Head.** We input the concat of $\hat{\boldsymbol{Z}}_i^j$ and $\hat{\boldsymbol{X}}_{i,S}^j$ into the LM backbone and obtain the representations $\boldsymbol{R}_i^j \in \mathbb{R}^{(M+B_i) \times D}$, which will be flattened and projected to the final results $\hat{\boldsymbol{Y}}_i^j \in \mathbb{R}^{1 \times F_i}$ by a linear layer.

**Personalized Strategy.** Time series across different domains could be substantially heterogeneous. Consequently, a generalized global model in FL may fail to capture the disparate temporal patterns and ultimately compromises the prediction performance. Inspired by [46], which indicates that diverse data may share common feature representations, we propose to learn a global encoder (i.e., *modality alignment*, *prompt adaption* and *LM backbone*) and domain-customized *prediction heads*. The underlying motivation is to strike a balance between generalization and personalization: (1) increasing the generalization of modality alignment and prompt adaption by access to cross-domain temporal patterns; (2) ensuring prediction results specific for certain domains by personalized heads. Since we keep the LM backbone intact, in each federating round, *only the parameters of modality alignment and prompt adaption are communicated*. The server performs aggregation by averaging strategy. The training strategy differs from Federated Averaging framework, where the parameters of encoder and decoder are both aggregated at the central server after local optimization.

### 3.3 Training Process

We denote $w_t^g$ as the global parameters of modality alignment and prompt adaption at the $t$-th federated round and $w_{i,t}^p$ as prediction head parameters of $i$ at the $t$-th round. We clarify that $(\boldsymbol{X}_i, \boldsymbol{Y}_i)$ here is reused to represent a training batch. $\hat{\boldsymbol{X}}_{i,S}, \hat{\boldsymbol{Z}}_i, \boldsymbol{R}_i$, and $\hat{\boldsymbol{Y}}_i$ denote the patch tokens, prompt tokens, representations and prediction results of such batch respectively. The training procedure of TIME-FFM is elaborated in Algorithm 1. **(1)** In the $t$-th federated round, the server distributes the global parameters $w_t^g$ (Line 8 and 9). **(2)** Each domain loads the global parameters and local head parameters to perform prediction following modality alignment, prompt adaption as well as representation obtaining from LM backbone (Line 12-15) and uploads $w_{t,i}^g$ to the server after optimization. **(3)** Finally, the server aggregates local updated parameters by averaging mechanism to obtain the fresh global parameters $w_{t+1}^g$ for the $(t+1)$-th round (Line 6).

## 4 Experiments

We comprehensively compare the proposed TIME-FFM with state-of-the-art models in FL or centralized settings, especially those by fine-tuning LM for time series forecasting. The numerical results demonstrate the effectiveness of TIME-FFM in time series forecasting. We employ GPT2

**Algorithm 1:** Training process of TIME-FFM.

---

**Input:** Global round number $T$, local epoch number $E$, initial global encoder parameters $w_0^g$, initial personalized head parameters $\{w_{i,0}^p\}$, local batch number $b_i$.

**Output:** Optimized global encoder parameters $w_T^g$, optimized parameters of personalized heads $\{w_{i,T}^p\}$.

1 **SERVEREXECUTE:**
2 **for** $t = 0, 1, \cdots, T-1$ **do**
3     **for** $i = 1, 2, \cdots, N$ ***in parallel* do**
4        $w_{t,i}^g \leftarrow$ LocalExecute $(i, w_t^g)$
5     $w_{t+1}^g = \frac{1}{N} \sum_{i \in [1,N]} w_{t,i}^g$
6 // *for local training*

7 **Function** LocalExecute($i, w_t^g$)**:**
8     $w_{t,i}^g \leftarrow w_t^g$
9     **for** $e = 1, 2, \cdots, E$ **do**
10        **for** $(\boldsymbol{X}_i, \boldsymbol{Y}_i)$ *in* $b_i$ *batches* **do**
11           $\hat{\boldsymbol{X}}_{i,S}, \hat{\boldsymbol{Z}}_i \leftarrow g(w_{t,i}^g; \boldsymbol{X}_i, \boldsymbol{E})$
12           $\boldsymbol{R}_i \leftarrow \text{LM}(\text{concat}(\hat{\boldsymbol{X}}_{i,S} || \hat{\boldsymbol{Z}}_i))$
13           $\hat{\boldsymbol{Y}}_i \leftarrow p(w_{i,t}^p; \boldsymbol{R}_n)$
14           $loss \leftarrow ||\boldsymbol{Y}_i - \hat{\boldsymbol{Y}}_i||_2^2$
15           Update $w_{t,i}^g$ and $w_{i,t}^p$ via gradient descent.
16     $w_{i,t+1}^p \leftarrow w_{i,t}^p$
17     **return** $w_{t,i}^g$

---

backbone of the first 6 layers as the default LM backbone and freeze all parameters. To guarantee a fair comparison, we adhere to the experimental configurations in [25].

**Baselines.** Our baselines cover a board collection of relevant methods, which can be categorised into 3 types: **TY1** (*federated fine-tuning methods*): FedIT [31], FedAdapter[H] [47, 41], and FedAdapter[P] [48, 41]; **TY2** (*across-dataset centralized methods*): UniTime [25], GPT4TS [23], and PatchTST [43]; [2] **TY3** (*dataset-specific centralized methods*): TimesNet [4], DLinear [49], FEDformer [50], Autoformer [10], and Informer [9]. We directly cite the results from [25] if applicable.

**Setups.** We evaluate on 8 benchmark datasets from various domains: ETTh1, ETTh2, ETTm1, ETTm2, Electricity, Weather, Exchange, and ILI, which have been widely adopted for evaluating time series forecasting performance. Each dataset corresponds to a FL participant. Detailed introduction of implementation and datasets can be found in Appendix A. We use Mean Square Error (MSE) and Mean Absolute Error (MAE) as the evaluation metrics.

### 4.1 Main Results

Main forecasting results are presented in Table 1. TIME-FFM consistently outperforms the other FL methods (in **TY1**) on all datasets, except ETTh2. Specifically, TIME-FFM can improve the performance gains over all datasets by 39.01% in terms of MSE, compared with the second best-performed FL method. Furthermore, the averaged prediction results of TIME-FFM are even superior to those of the centralized models. When compared with UniTime, the recently-proposed centralized unified model for cross-domain time series forecasting, TIME-FFM can provide more performance gains, which underscores the effectiveness of the proposed cross-modality adaption modules and personalized approach.

### 4.2 Few-Shot Forecasting

Given the remarkable few-shot learning performance of LMs, we evaluate whether TIME-FFM can retain such capability for time series forecasting. In this section, we compare the prediction performance across **TY1** and **TY2** in few-shot settings with 10% and 5% time steps adopted as training samples, which is in line with the setups in [23, 22].

Main results of 10% and 5% few-shot forecasting are presented in Table 2 and 3 respectively. TIME-FFM outperforms the other FL methods and even achieves comparable performance in contrast to the centralized methods, which further underscores that TIME-FFM inherits the few-shot capability of LMs and promises proficient FM for time series forecasting. Specifically, TIME-FFM outperforms the centralized methods in the realm of 5% few-shot learning, with 20% reduction in averaged MSE w.r.t UniTime. Interestingly, for all methods except UniTime, results in 10% few-shot learning are worse than those in 5% few-shot learning. We deduce that the pretrained LM is fully-tuned in

---

[2]Here we modify the original GPT4TS and PatchTST as per [25].

Table 1: Forecasting performance comparisons. All results are averaged over four prediction windows, i.e., $F_i \in \{24, 36, 48, 60\}$ for ILI and $\{96, 192, 336, 720\}$ for others. Yellow : the best in **TY1**; Blue : the second best in **TY1**. Underline: the best over all types; **Bold**: the second best over all types. Full results are presented in Table 13.

| Type | TY1 | | | | | | | | TY2 | | | | | | TY3 | | | | | | | | | |
|---|---|---|---|---|---|---|---|---|---|---|---|---|---|---|---|---|---|---|---|---|---|---|---|---|
| Method | TIME-FFM | | FedIT | | FedAdapter$^H$ | | FedAdapter$^P$ | | UniTime | | GPT4TS | | PatchTST | | TimesNet | | DLinear | | FEDformer | | Autoformer | | Informer | |
| Metric | MSE | MAE | MSE | MAE | MSE | MAE | MSE | MAE | MSE | MAE | MSE | MAE | MSE | MAE | MSE | MAE | MSE | MAE | MSE | MAE | MSE | MAE | MSE | MAE |
| ETTh1 | 0.442 | 0.434 | 0.481 | 0.461 | 0.488 | 0.467 | 0.503 | 0.479 | **0.442** | **0.448** | 0.502 | 0.461 | 0.472 | 0.451 | 0.458 | 0.450 | 0.456 | 0.452 | 0.440 | 0.460 | 0.496 | 0.487 | 1.040 | 0.795 |
| ETTh2 | 0.382 | 0.406 | **0.374** | **0.396** | 0.373 | 0.398 | 0.380 | 0.403 | 0.378 | 0.403 | 0.386 | 0.406 | 0.398 | 0.416 | 0.414 | 0.427 | 0.559 | 0.515 | 0.437 | 0.449 | 0.450 | 0.459 | 4.431 | 1.729 |
| ETTm1 | 0.399 | **0.402** | 0.644 | 0.517 | 0.643 | 0.511 | 0.640 | 0.516 | **0.385** | **0.399** | 0.551 | 0.483 | 0.971 | 0.629 | 0.383 | 0.406 | 0.403 | 0.407 | 0.448 | 0.452 | 0.588 | 0.517 | 0.961 | 0.734 |
| ETTm2 | **0.286** | **0.332** | 0.297 | 0.341 | 0.295 | 0.340 | 0.298 | 0.342 | 0.293 | 0.334 | 0.321 | 0.356 | 0.340 | 0.373 | **0.291** | **0.322** | 0.350 | 0.401 | 0.305 | 0.349 | 0.327 | 0.371 | 1.410 | 0.810 |
| Electricity | 0.216 | **0.299** | 0.390 | 0.478 | 0.408 | 0.489 | 0.334 | 0.420 | 0.216 | 0.305 | 0.251 | 0.338 | 0.221 | 0.311 | 0.193 | 0.295 | 0.212 | 0.300 | 0.214 | 0.327 | 0.227 | 0.338 | 0.311 | 0.397 |
| Weather | 0.270 | 0.288 | 0.282 | 0.310 | 0.282 | 0.308 | 0.287 | 0.309 | **0.253** | **0.276** | 0.293 | 0.309 | 0.304 | 0.323 | **0.259** | **0.287** | 0.265 | 0.317 | 0.309 | 0.360 | 0.338 | 0.382 | 0.634 | 0.548 |
| Exchange | **0.338** | 0.391 | 0.389 | 0.423 | 0.382 | 0.419 | 0.380 | 0.417 | 0.364 | **0.404** | 0.421 | 0.446 | 0.411 | 0.444 | 0.416 | 0.443 | **0.354** | 0.414 | 0.519 | 0.500 | 0.613 | 0.539 | 1.550 | 0.998 |
| ILI | **2.107** | 0.924 | 4.423 | 1.448 | 5.247 | 1.621 | 5.251 | 1.600 | **2.137** | **0.929** | 3.678 | 1.372 | 4.210 | 1.480 | 2.139 | 0.931 | 2.616 | 1.090 | 2.847 | 1.144 | 3.006 | 1.161 | 5.137 | 1.544 |
| Average | 0.555 | 0.434 | 0.910 | 0.547 | 1.015 | 0.569 | 1.009 | 0.561 | **0.559** | **0.437** | 0.800 | 0.521 | 0.916 | 0.553 | 0.569 | 0.445 | 0.652 | 0.487 | 0.690 | 0.505 | 0.756 | 0.532 | 1.934 | 0.944 |
| 1$^{st}$ Count | 8 | | 1 | | 1 | | 0 | | 3 | | 0 | | 0 | | 4 | | 0 | | 1 | | 0 | | 0 | |

UniTime and fewer training samples fail to support optimizing masses of parameters. While in the other methods, the pretrained LMs are frozen or fine-tuned, which can retain the original reasoning capability of LMs even with fewer training instances.

Table 2: 10% few-shot forecasting results. All results are averaged across four prediction windows, i.e., $F_i \in \{96, 192, 336, 720\}$. Yellow : the best in **TY1**; Blue : the second best in **TY1**. Underline: the best over both types; **Bold**: the second best over both types. Full results are presented in Table 14.

| Type | TY1 | | | | | | | | TY2 | | | | | |
|---|---|---|---|---|---|---|---|---|---|---|---|---|---|---|
| Method | TIME-FFM | | FedLoRA | | FedAdapter$^H$ | | FedAdapter$^P$ | | UniTime | | GPT4TS | | PatchTST | |
| Metric | MSE | MAE | MSE | MAE | MSE | MAE | MSE | MAE | MSE | MAE | MSE | MAE | MSE | MAE |
| ETTm1 | **0.593** | **0.500** | 0.637 | 0.506 | 0.672 | 0.539 | 0.697 | 0.543 | 0.589 | 0.494 | 0.638 | 0.501 | 1.071 | 0.662 |
| ETTm2 | 0.294 | 0.335 | 0.297 | 0.340 | 0.298 | 0.341 | 0.298 | 0.343 | 0.299 | 0.338 | **0.295** | **0.336** | 0.348 | 0.378 |
| Electricity | 0.266 | 0.344 | 0.275 | 0.363 | 0.421 | 0.489 | 0.408 | 0.486 | **0.254** | **0.342** | 0.251 | 0.334 | 0.362 | 0.429 |
| Weather | 0.288 | 0.314 | 0.296 | 0.320 | **0.284** | **0.311** | 0.287 | 0.315 | 0.272 | 0.299 | 0.300 | 0.322 | 0.297 | 0.316 |
| Exchange | 0.230 | 0.336 | 0.238 | 0.339 | **0.227** | 0.334 | 0.230 | 0.335 | 0.220 | 0.331 | 0.242 | 0.344 | 0.220 | 0.330 |
| Average | 0.334 | 0.366 | 0.349 | 0.374 | 0.380 | 0.403 | 0.384 | 0.404 | 0.327 | 0.361 | 0.345 | 0.367 | 0.459 | 0.423 |
| 1$^{st}$ Count | 2 | | 0 | | 0 | | 0 | | 7 | | 2 | | 2 | |

Table 3: 5% few-shot forecasting results. All results are averaged across four prediction windows, i.e., $F_i \in \{96, 192, 336, 720\}$. Yellow : the best in **TY1**; Blue : the second best in **TY1**. Underline: the best over both types; **Bold**: the second best over both types. Full results are presented in Table 16.

| Type | TY1 | | | | | | | | TY2 | | | | | |
|---|---|---|---|---|---|---|---|---|---|---|---|---|---|---|
| Method | TIME-FFM | | FedLoRA | | FedAdapter$^H$ | | FedAdapter$^P$ | | UniTime | | GPT4TS | | PatchTST | |
| Metric | MSE | MAE | MSE | MAE | MSE | MAE | MSE | MAE | MSE | MAE | MSE | MAE | MSE | MAE |
| ETTm1 | 0.567 | 0.491 | 0.606 | **0.494** | 0.650 | 0.526 | 0.636 | 0.519 | 0.713 | 0.558 | 0.631 | 0.522 | **0.591** | 0.497 |
| ETTm2 | 0.293 | 0.333 | 0.298 | 0.339 | 0.298 | 0.339 | **0.296** | **0.338** | 0.313 | 0.350 | 0.298 | 0.339 | 0.299 | 0.339 |
| Electricity | 0.324 | 0.403 | 0.339 | 0.420 | 0.333 | 0.411 | 0.333 | 0.409 | **0.298** | **0.387** | 0.273 | 0.355 | 0.309 | 0.391 |
| Weather | 0.292 | 0.317 | 0.303 | 0.325 | 0.292 | 0.317 | 0.300 | 0.322 | 0.288 | 0.313 | **0.288** | **0.314** | 0.301 | 0.324 |
| Exchange | 0.167 | 0.289 | 0.171 | 0.291 | 0.166 | **0.288** | 0.166 | 0.287 | 0.442 | 0.493 | 0.168 | 0.290 | 0.171 | 0.293 |
| Average | 0.329 | 0.367 | 0.344 | 0.374 | 0.348 | 0.376 | 0.346 | 0.375 | 0.411 | 0.420 | **0.332** | 0.364 | 0.334 | 0.369 |
| 1$^{st}$ Count | 5 | | 0 | | 1 | | 2 | | 2 | | 4 | | 0 | |

## 4.3 Zero-Shot Forecasting

Given that language FMs are effective zero-shot forecasters, we evaluate the zero-shot learning capability of TIME-FFM, which is essential for a FM. We adhere to the zero-shot learning settings in [25], where we first train TIME-FFM on ETTh1, ETTm1, and ETTm2, and then evaluate the zero-shot testing performance on ETTh2, Electricity, and Weather.

Since ETTh2 hails from the same domain of ETTh1, we directly reuse the *local parameters* (including both encoder and head) of ETTh1 for inferring ETTh2. For the other two target datasets from different domains of the source datasets, we successively reuse local parameters of the three source datasets to perform zero-shot testing. The results presented in Table 15 show that local parameters of ETTh1 excel on both target datasets. Hence, we adopt the model parameters of ETTh1 for zero-shot testing on Electricity and Weather. For other methods in **TY1**, we train an optimized global model on ETTh1, ETTm1, and ETTm2, and then adopt the obtained global model to conduct zero-shot testing on ETTh2, Electricity, and Weather. The comparison in zero-shot forecasting is presented in Table 4. TIME-FFM consistently ensures significant performance gains on all three datasets, with prediction MSE decreasing by 13.9% w.r.t the second best. It is remarkable that the centralized unified model UniTime exhibits inferior zero-shot testing performance compared to TIME-FFM. We attribute the performance gains of TIME-FFM to the valid knowledge transferability across domains.

Table 4: Zero-shot forecasting results. All results are averaged across four prediction windows, i.e., $F_i \in \{96, 192, 336, 720\}$. Yellow : the best in **TY1**; Blue : the second best in **TY1**. **Underline**: the best over both types; **Bold**: the second best over both types. Full results are presented in Table 17.

| Type | | | TY1 | | | | | | | | TY2 | | | | |
|---|---|---|---|---|---|---|---|---|---|---|---|---|---|---|---|
| Method | | TIME-FFM | | FedIT | | FedAdapter[H] | | FedAdapter[P] | | UniTime | | GPT4TS | | PatchTST | |
| Metric | MSE | MAE | MSE | MAE | MSE | MAE | MSE | MAE | MSE | MAE | MSE | MAE | MSE | MAE |
| ETTh2 | 0.373 | 0.399 | 0.387 | 0.407 | 0.388 | 0.408 | 0.387 | 0.407 | 0.388 | 0.409 | 0.397 | 0.418 | 0.421 | 0.429 |
| Electricity | 0.265 | 0.343 | 0.398 | 0.470 | 0.401 | 0.474 | 0.409 | 0.482 | 0.436 | 0.500 | 0.462 | 0.526 | 0.534 | 0.565 |
| Weather | 0.291 | 0.318 | 0.295 | 0.319 | 0.302 | 0.324 | 0.302 | 0.324 | 0.301 | 0.320 | 0.322 | 0.339 | 0.327 | 0.339 |
| Average | 0.310 | 0.353 | 0.360 | 0.399 | 0.364 | 0.402 | 0.366 | 0.404 | 0.375 | 0.410 | 0.394 | 0.428 | 0.427 | 0.444 |

Table 5: Ablation studies of TIME-FFM on ETTh1 and ILI datasets with $F_i \in \{336, 720\}$ and $F_i \in \{48, 60\}$ respectively. **Bold**: the best.

| Foreccasting Task | | ETTh1-336 | | ETTh1-720 | | ILI-48 | | ILI-60 | |
|---|---|---|---|---|---|---|---|---|---|
| Metric | | MSE | MAE | MSE | MAE | MSE | MAE | MSE | MAE |
| **A.1** TIME-FFM | | **0.480** | **0.449** | **0.462** | **0.456** | **1.953** | **0.894** | **1.976** | **0.916** |
| **A.2** w/o Prompt Adaption | | 0.495 | 0.450 | 0.496 | 0.471 | 2.222 | 0.947 | 2.118 | 0.952 |
| **A.3** w/ Instructions | | 0.487 | 0.457 | 0.465 | 0.465 | 2.109 | 0.953 | 2.170 | 0.977 |
| **A.4** w/o Personalized Head | | 0.537 | 0.471 | 0.526 | 0.480 | 4.953 | 1.591 | 4.068 | 1.450 |
| **A.5** w/o All | | 0.562 | 0.498 | 0.523 | 0.495 | 8.153 | 2.037 | 6.509 | 1.804 |
| **A.6** TIME-FFM-D | | 0.499 | 0.450 | 0.503 | 0.472 | 2.453 | 1.022 | 2.427 | 1.026 |

## 4.4 Model Analysis

**Model Ablation.** We conduct ablation studies on five variants of TIME-FFM and the corresponding results are presented in Table 5 (**A.1**-**A.6**). Thereinto, TIME-FFM-D represents the distirbuted version of TIME-FFM, which ablates the aggregation process. The results demonstrate that ablating either components will compromise the forecasting performance. We have the following key observations: **(1)** The prompt tokens can bootstrap the LM for target domains. The absence of prompt adaption (**A.2**) will affect the forecasting performance. When employing instructions in [25] as prompts, **A.3** is inferior to TIME-FFM, which underscores the efficacy of prompt adaption. **(2)** The ablation of personalized heads (**A.4**) will hurt the performance most. In **A.4**, a global prediction head is learned for all domains, hardly ensuring the personalization for cross-domain heterogeneous data. **(3)** In **A.6**, the common temporal knowledge fails to be shared among domains, which makes poorer generalization of cross-modality adaption modules, thus yielding inferior performance. This underscores the significance of building a unified model for cross-domain traffic series forecasting.

Table 6: Ablation studies of LM on ETTh1 and Weather datasets with $F_i \in \{96, 192\}$ and $F_i \in \{336, 720\}$ respectively. **Bold**: the best.

| Forecasting Task | ETTh1-96 | | ETTh1-192 | | Weather-336 | | Weather-720 | |
|---|---|---|---|---|---|---|---|---|
| Metric | MSE | MAE | MSE | MAE | MSE | MAE | MSE | MAE |
| **B.1** Freeze (**Default**) | 0.422 | 0.412 | 0.473 | 0.439 | 0.295 | 0.308 | 0.367 | 0.354 |
| **B.2** FPT | 0.396 | 0.409 | 0.450 | 0.441 | 0.290 | 0.305 | 0.363 | 0.352 |
| **B.3** Full | **0.394** | **0.403** | **0.448** | **0.431** | **0.287** | **0.305** | **0.360** | **0.351** |
| **C.1** GPT2 (6) (**Default**) | 0.422 | 0.412 | 0.473 | 0.439 | 0.295 | 0.308 | 0.367 | 0.354 |
| **C.2** GPT2 (12) | **0.406** | **0.409** | **0.456** | **0.436** | **0.294** | **0.307** | **0.367** | **0.353** |

Table 7: Efficiency analysis of TIME-FFM on ETTh1 dataset.

| Method | Training Param. (M) | Total Param. (M) | Training Param. PCT. (%) | Training Time (s/iter) | Comm. Param. (M) |
|---|---|---|---|---|---|
| FedLoRa | 8.543 | 90.456 | 9.445 | 0.048 | 8.543 |
| FedAdapter[H] | 47.998 | 90.945 | 52.777 | 0.062 | 47.998 |
| FedAdapter[P] | 47.550 | 90.498 | 52.543 | 0.046 | 47.550 |
| TIME-FFM | 8.138 | 90.050 | 9.037 | 0.088 | 6.811 |
| GPT (12) | 8.138 | 132.578 | 6.138 | 0.156 | 6.811 |

**Language Model Variants.** We investigate the variants of LM, in terms of optimization modes (**B.1**-**B.3**) and backbone layers (**C.1** and **C.2**). Here we train all variants on seven datasets except Electricity, due to GPU memory limitation. In **B.3**, the backbones of LM are full-tuned. While in **B.2**, we only tune the positional embeddings and layer normalization components of the backbone [23]. Table 6 shows that **B.3** performs best, followed by **B.2** and **B.1**. We argue that the performance remains comparable when we freeze all backbone parameters. This demonstrates that LMs are capable in processing time series tokens by effectively modality alignment. In **C.1** and **C.2**, 6 and 12 backbone layers are adopted. The results shows that more backbone layers ensure better performance, which indicates the scaling laws of LMs retain in TIME-FFM for time series forecasting [51, 22].

**Model Efficiency.** Table 7 demonstrates that TIME-FFM can reduce the training parameter quantity and communication overhead with insignificant increase in training time. Moreover, *the number of training parameters and communication parameters keeps intact*, regardless of backbone layers.

**Case Study.** We provide a case study on prompt adaption in Figure 3. **(a)** shows the attention scores between 6 patch tokens and 100 text prototypes for 8 heads on ETTh1 dataset. For each head, only a small set of text prototypes (columns) have remarkable scores, which indicates that each patch token is only related to limited pretrained word embeddings and dynamically prompt adaption is promising. **(b)-(d)** show top $M$ prototypes of 8 heads on ETTh1, Electricity, and ILI respectively. Darker colors correspond to text prototypes with higher attention scores. From these three subplots,

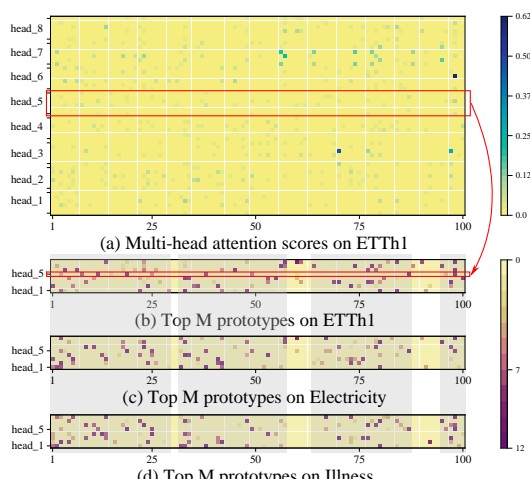

(a) Multi-head attention scores on ETTh1

(b) Top M prototypes on ETTh1

(c) Top M prototypes on Electricity

(d) Top M prototypes on Illness

Figure 3: A showcase of prompt adaption.

we have the following key observations: (1) different datasets correspond to variant text prototypes; (2) the distribution of text prototypes on different datasets has commonality, i.e., gathering in shadow areas. These observations indicate the global prompt adaption module has great generalization for diverse datasets and simultaneously ensures personalization across various domains.

## 5 Conclusion and Discussion

In this paper, we propose the first federated foundation model for time series forecasting, with adaptively generating domain-specific prompts and tackling time series heterogeneity for general-

purpose learning and personalized prediction. Specifically, given the differentiation of dimensionality and horizon, we introduce the modality alignment module encompassing the channel-independent and patching techniques, which may follow the track of GPT4TS and Time-LLM. For bootstrapping the pre-trained GPT2 backbone for cross-domain time series reasoning, we propose to adaptively construct prompts from how to understand patch tokens, rather than from rigid domain instructions. Due to cross-domain time series heterogeneity, we devise a personalized federated strategy, with global encoder and personalized prediction heads.

**Rationale of TIME-FFM.** Compared with the modality of text, time series is more domain-specific and copyright-sensitive, i.e., private knowledge may be inferred from historical time series readings, especially in finance and healthcare domain. Hence, it is of great significance to take data privacy into account when constructing time series foundation models. Moreover, a multitude of public data cannot even be adopted for pre-training foundation models due to data license restriction, such as Kaggle public datasets. Hence, our work uniquely bridges the gap between foundation models and federated learning, which not only enhances the privacy and applicability of foundation models in sensitive domains but also opens up new avenues for leveraging rich, yet previously inaccessible, time series data for advanced predictive analytics, addressing a crucial need in this field.

**Limitations and Future Works.** We recognize some limitations of our work: the training time is increased compared with the **TY1** and the performance in some case is suboptimal. In the future work, we will explore more effective and efficient modality alignment strategies. Moreover, further researches will investigate the correspondence between patch embeddings and word embeddings to explore whether time series data can be seamlessly "translated" into natural language.

## Acknowledgments and Disclosure of Funding

This work is mainly supported by the National Natural Science Foundation of China (No. 62402414). This work is also supported by the Guangzhou-HKUST(GZ) Joint Funding Program (No. 2024A03J0620), Guangzhou Municipal Science and Technology Project (No. 2023A03J0011), the Guangzhou Industrial Information and Intelligent Key Laboratory Project (No. 2024A03J0628), and a grant from State Key Laboratory of Resources and Environmental Information System, and Guangdong Provincial Key Lab of Integrated Communication, Sensing and Computation for Ubiquitous Internet of Things (No. 2023B1212010007).

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

# A    Experimental Details

**Implementation.** The Adam optimizer with the initial learning rate of $10^{-4}$ is adopted in the training process. The lookback window $L_i$ is set to 36 for the ILI dataset, and 96 for the others. The future prediction window $F_i$ is set to $\{24, 36, 48, 60\}$ for the ILI dataset, and $\{96, 192, 336, 720\}$ for other ones. We adopt the pretrained GPT2-backbone of the first 6 layers as the LM encoder. The local epoch $E$ is set to 1 for all domains. The global round number $T$ is set to 100. $V'$, $M$, $P$ and $H$ are set to 100, 12, 16, and 8 respectively for all domains. $S_i$ is set to 4 for the ILI dataset, and 16 for other ones. In each round, we calculate the averaged values of validation loss. The round with lowest validation value serves as the optimal round, and then the corresponding model is used for test. All models are implemented on PyTorch with all experiments conducted on NVIDIA A100-80G GPUs.

Table 8: Detailed descriptions of datasets. The dataset size is organized in (training, validation, test).

| Dataset | $c_i$ | Dataset Size | Batch Size | OverSampling Times | Frequency | Application Domain |
|---|---|---|---|---|---|---|
| ETTh1 | 7 | (8545, 2881, 2881) | 32 | - | 1 hour | Electrical Asset Monitoring |
| ETTh2 | 7 | (8545, 2881, 2881) | 32 | - | 1 hour | Electrical Asset Monitoring |
| ETTm1 | 7 | (34465, 11521, 11521) | 64 | - | 15 minutes | Electrical Asset Monitoring |
| ETTm2 | 7 | (34465, 11521, 11521) | 64 | - | 15 minutes | Electrical Asset Monitoring |
| Electricity | 321 | (18317, 2633, 5261) | 24 | - | 1 hour | Energy Consumption |
| Weather | 21 | (36792, 5271, 10540) | 64 | - | 10 minutes | Weather Forecasting |
| Exchange | 8 | (5120, 665, 1422) | 24 | - | 1 day | International Trade |
| ILI | 7 | (617, 74, 170) | 16 | 12 | 1 week | Illness Monitoring |

**Training Configurations.** The experimental evaluations are conducted on 8 real-world benchmark datasets which include 5 domains. We present the detailed description of these datasets in Table 8. For fair comparison, we perform batch division and oversampling as per [25]. In each federated round, we do not train local models with all training samples, considering large quantity of training samples. Instead, we proportionately calculate the number of batches for each domain in the following steps. (1) We calculate the summation of training batches over all datsets before oversampling. (2) We count training times of each domain after oversampling, i.e., 13 for ILI and 1 for the others, and then we perform normalization to obtain a batch ratio for each domain, i.e., 0.65 for ILI and 0.05 for the others. (3) we can obtain the number of training batches for each domain (denoted as $b_i$) by multiply the summation (in (1)) and ratios (in (2)) respectively. Actually, for ILI the value is higher than the number of training batches, while the opposite is true for the others. In each round, each local model is trained with training batches sequentially until $b_i$ is reached.

We evaluate the effectiveness of oversampling strategy in TIME-FFM and present the results in Table 9. "w/o OverSampling" represents each local model is trained with all local batches in each FL round. We attribute the performance gains in TIME-FFM to it that the introduction of oversampling strategy can balance the contribution to the global knowledge. For ILI, despite data sparsity, its local knowledge can be augmented in the global encoder. We observe that such local knowledge can enhance forecasting for not only ILI itself but also the other domains.

Table 9: Effectiveness evaluation of oversampling. All results are averaged over four prediction windows, i.e., $F_i \in \{24, 36, 48, 60\}$ for ILI and $\{96, 192, 336, 720\}$ for others. **Bold**: Better.

| Datasets | ETTh1 | | ETTh2 | | ETTm1 | | ETTm2 | | Electricity | | Weather | | Exchange | | ILI | |
|---|---|---|---|---|---|---|---|---|---|---|---|---|---|---|---|---|
| Metrics | MSE | MAE | MSE | MAE | MSE | MAE | MSE | MAE | MSE | MAE | MSE | MAE | MSE | MAE | MSE | MAE |
| TIME-FFM | **0.442** | **0.434** | **0.382** | **0.406** | **0.399** | **0.402** | **0.286** | **0.332** | 0.216 | 0.299 | **0.270** | **0.288** | **0.338** | **0.391** | **2.107** | **0.924** |
| w/o OverSampling | 0.456 | 0.445 | 0.396 | 0.414 | 0.405 | 0.410 | 0.300 | 0.341 | **0.212** | **0.295** | 0.272 | 0.289 | 0.345 | 0.393 | 2.364 | 0.989 |

# B    Hyperparameter Sensitivity

In this section, we conduct hyperparameter investigation of 3 important hyperparameters, i.e., the number of text prototypes $V'$, the number of prompt tokens $M$, and the number of self-attention heads $H$. Figure 4 shows prediction performance on ILI dataset with the variation of the 3 hyperparameters respectively. We have the key observations as follows: (1) When the value of $V'$ is lower, word

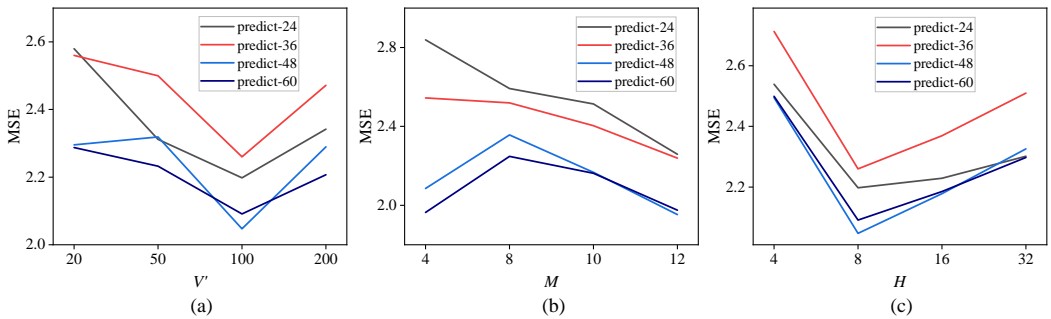

Figure 4: Hyperparameter sensitivity studies on ILI dataset.

embeddings are projected into less text prototypes. Each prototype will contain both relevant and irrelevant knowledge, which will affect the accuracy of prompt adaption. When text prototypes are more, a stable number of prompt tokens will not cover all relevant knowledge. Hence lower or higher values of $V'$ will yield subpar performance. (2) Fewer prompt tokens may not fully cover the useful knowledge. Hence, the best performance is achieved when $M$ is equal to 12. (3) Increasing the number of attention heads cannot always promise better performance because more heads may break the semantic integrity of text prototypes and patch embeddings.

## C Additional Results

We compare forecasting performance with PatchTST-FL and DLinear-FL, the federated version of PatchTST and DLinear. As is presented in Table 10, TIME-FFM consistently outperforms the two novel federated methods on all datasets.

Table 10: Performance comparison with PatchTST-FL and DLinear-FL. **Bold**: the best.

| Method | TIME-FFM | | PatchTST-FL | | DLinear-FL | |
|---|---|---|---|---|---|---|
| Metric | MSE | MAE | MSE | MAE | MSE | MAE |
| ETTh1 | **0.442** | **0.434** | 0.534 | 0.496 | 0.565 | 0.545 |
| ETTh2 | **0.382** | **0.406** | 0.399 | 0.415 | 1.040 | 0.738 |
| ETTm1 | **0.399** | **0.402** | 0.752 | 0.573 | 0.783 | 0.627 |
| ETTm2 | **0.286** | **0.332** | 0.318 | 0.357 | 0.987 | 0.730 |
| Electricity | **0.216** | **0.299** | 0.457 | 0.523 | 0.363 | 0.452 |
| Weather | **0.270** | **0.288** | 0.288 | 0.317 | 0.339 | 0.402 |
| Exchange | **0.338** | **0.391** | 0.404 | 0.440 | 0.830 | 0.723 |
| Average | **0.333** | **0.364** | 0.450 | 0.446 | 0.701 | 0.602 |

We further compare the forecasting performance with three baselines, i.e., iTransformer [52], N-BEATS [53], and Crossformer [54]. These three baselines can be categorized into **TY3**. The numerical results are presented in Table 11. We have the key observation that TIME-FFM, though trained in federated paradigm, can outperform these three centralized methods.

Table 11: Performance comparison with iTransformer, N-BEATS, and Crossformer. **Bold**: the best.

| Method | TIME-FFM | | iTransformer | | N-BEATS | | Crossformer | |
|---|---|---|---|---|---|---|---|---|
| Metric | MSE | MAE | MSE | MAE | MSE | MAE | MSE | MAE |
| ETTm2 | **0.286** | **0.332** | 0.288 | 0.332 | 0.294 | 0.345 | 0.757 | 0.610 |
| Weather | 0.270 | 0.288 | **0.258** | **0.278** | 0.263 | 0.282 | 0.259 | 0.315 |
| Exchange | **0.338** | **0.391** | 0.360 | 0.403 | 0.481 | 0.455 | 0.940 | 0.707 |
| Average | **0.298** | **0.337** | 0.302 | 0.338 | 0.346 | 0.361 | 0.652 | 0.544 |

Some researches delve into training a foundation model from scratch based on the collected time series datasets [55, 32, 56]. We compare our proposed federated foundation model with Moirai [32] and MOMENT [56] in Table 12. Notably, TIME-FFM achieves comparable performance with the two foundation models which are pre-trained firstly on large-scale time series archive.

Table 12: Performance comparison with MOIRAI and Moment. **Bold**: the best.

| Method | TIME-FFM | | Moirai | | MOMENT | |
|---|---|---|---|---|---|---|
| Metric | MSE | MAE | MSE | MAE | MSE | MAE |
| ETTh1 | 0.442 | 0.434 | **0.400** | **0.424** | 0.418 | 0.436 |
| ETTh2 | 0.382 | 0.406 | **0.341** | **0.379** | 0.352 | 0.395 |
| ETTm1 | 0.399 | 0.402 | 0.448 | 0.409 | **0.344** | **0.379** |
| ETTm2 | 0.286 | 0.332 | 0.300 | 0.341 | **0.259** | **0.318** |
| Electricity | 0.216 | 0.299 | 0.233 | 0.320 | **0.165** | **0.260** |
| Weather | 0.270 | 0.288 | 0.242 | **0.267** | **0.228** | 0.270 |

## D  Full Results

Full results of forecasting performance comparison on 8 time series benchmarks are presented in Table 13. TIME-FFM exhibits SOTA performance in *32 out of 42 instances*, which demonstrates the effectiveness of the cross-modality adaption module, i.e., modality alignment and prompt adaption, as well as the personalized prediction heads.

Our complete results of performance comparison in 10% and 5% few-shot settings are presented in Table 14 and 16 respectively. In both settings, TIME-FFM outperforms the other FL methods in **TY1**. In the setting of 10% few-shot forecasting, TIME-FFM achieves comparable performance against methods in **TY2**. In the setting of 5% few-shot learning, TIME-FFM attains SOTA performance on *20 out of 48 instances* across five time series benchmarks. The results underscore that TIME-FFM promises effective few-shot forecaster.

## E  Error Bars

We conduct the experiments of **TY1** for three times and report the mean values and standard deviations in Table 18. The results demonstrate the superiority of our proposed TIME-FFM, which agrees with Table 1.

## F  Border Impacts

In this paper, we propose to build a foundation model for time series forecasting hinging on the impressive capability of pretrained language models for sequence tokens reasoning. The promising advantages are two folds: (1) Data owners do not need to share the access to the private data samples which mitigates the privacy concerns and cater for data protection regulations (say GDPR). (2) The problem of "data island" can be tackled, which makes it possible to generate satisfactory performance in spite of data scarcity. To the best of our knowledge, our research do not have obvious negative social impacts.

Table 13: Full results of forecasting performance comparisons. Yellow : the best in **TY1**; Blue : the second best in **TY1**. Underline: the best over all types; **Bold**: the second best over all types.

| Type | | TY1 | | | | | | TY2 | | | | | | TY3 | | | | | | | | | |
|---|---|---|---|---|---|---|---|---|---|---|---|---|---|---|---|---|---|---|---|---|---|---|---|
| Method | | TIME-FFM | | FedIT | | FedAdapterH | | FedAdapterP | | UniTime | | GPT4TS | | PatchTST | | TimesNet | | DLinear | | FEDformer | | Autoformer | | Informer | |
| Metric | | MSE | MAE | MSE | MAE | MSE | MAE | MSE | MAE | MSE | MAE | MSE | MAE | MSE | MAE | MSE | MAE | MSE | MAE | MSE | MAE | MSE | MAE | MSE MAE |
| ETTh1 | 96 | 0.385 | **0.400** | 0.446 | 0.436 | 0.455 | 0.441 | 0.447 | 0.441 | 0.397 | 0.418 | 0.449 | 0.424 | 0.409 | 0.403 | **0.384** | **0.402** | 0.386 | **0.400** | 0.376 | 0.419 | 0.449 | 0.459 | 0.865 0.713 |
| | 192 | 0.439 | 0.430 | 0.480 | 0.454 | 0.486 | 0.459 | 0.481 | 0.461 | **0.434** | 0.439 | 0.503 | 0.453 | 0.467 | 0.444 | 0.436 | **0.429** | 0.437 | 0.432 | 0.420 | 0.448 | 0.500 | 0.482 | 1.008 0.792 |
| | 336 | 0.480 | **0.449** | 0.508 | 0.471 | 0.514 | 0.476 | 0.528 | 0.489 | **0.468** | 0.457 | 0.540 | 0.477 | 0.509 | 0.472 | 0.491 | 0.469 | 0.481 | 0.459 | 0.459 | 0.465 | 0.521 | 0.496 | 1.107 0.809 |
| | 720 | **0.462** | **0.456** | 0.488 | 0.484 | 0.496 | 0.491 | 0.554 | 0.524 | 0.469 | 0.477 | 0.515 | 0.489 | 0.503 | 0.485 | 0.521 | 0.500 | 0.519 | 0.516 | 0.506 | 0.507 | 0.514 | 0.512 | 1.181 0.865 |
| | AVG | **0.442** | **0.434** | 0.481 | 0.461 | 0.488 | 0.467 | 0.503 | 0.479 | **0.442** | 0.448 | 0.502 | 0.461 | 0.472 | 0.451 | 0.458 | 0.450 | 0.456 | 0.452 | 0.440 | 0.460 | 0.496 | 0.487 | 1.040 0.795 |
| ETTh2 | 96 | 0.301 | 0.351 | 0.286 | 0.336 | 0.289 | 0.340 | 0.298 | 0.346 | **0.296** | 0.345 | 0.303 | 0.349 | 0.314 | 0.361 | 0.340 | 0.374 | 0.333 | 0.387 | 0.358 | 0.397 | 0.346 | 0.388 | 3.755 1.525 |
| | 192 | 0.378 | 0.397 | 0.373 | 0.387 | 0.375 | 0.390 | 0.379 | 0.394 | **0.374** | 0.394 | 0.391 | 0.399 | 0.407 | 0.411 | 0.402 | 0.414 | 0.477 | 0.476 | 0.429 | 0.439 | 0.456 | 0.452 | 5.602 1.931 |
| | 336 | 0.422 | 0.431 | 0.419 | 0.423 | 0.413 | 0.424 | 0.418 | 0.427 | **0.415** | 0.427 | 0.422 | 0.428 | 0.437 | 0.443 | 0.452 | 0.452 | 0.594 | 0.541 | 0.496 | 0.487 | 0.482 | 0.486 | 4.721 1.835 |
| | 720 | 0.427 | 0.444 | 0.418 | 0.436 | 0.416 | 0.438 | 0.426 | 0.443 | 0.425 | 0.444 | 0.429 | 0.449 | 0.434 | 0.448 | 0.462 | 0.468 | 0.831 | 0.657 | 0.463 | 0.474 | 0.515 | 0.511 | 3.647 1.625 |
| | AVG | 0.382 | 0.406 | 0.374 | 0.396 | 0.373 | 0.398 | 0.380 | 0.403 | 0.378 | 0.403 | 0.386 | 0.406 | 0.398 | 0.416 | 0.414 | 0.427 | 0.559 | 0.515 | 0.437 | 0.449 | 0.450 | 0.459 | 4.431 1.729 |
| ETTm1 | 96 | 0.336 | 0.369 | 0.609 | 0.495 | 0.610 | 0.489 | 0.618 | 0.498 | 0.322 | 0.363 | 0.509 | 0.463 | 0.927 | 0.604 | 0.338 | 0.375 | 0.345 | 0.372 | 0.379 | 0.419 | 0.505 | 0.475 | 0.672 0.571 |
| | 192 | 0.378 | 0.389 | 0.639 | 0.512 | 0.641 | 0.507 | 0.639 | 0.508 | 0.366 | 0.387 | 0.537 | 0.476 | 0.964 | 0.620 | **0.374** | **0.387** | 0.380 | 0.389 | 0.426 | 0.441 | 0.553 | 0.496 | 0.795 0.669 |
| | 336 | 0.411 | 0.410 | 0.653 | 0.521 | 0.648 | 0.515 | 0.637 | 0.515 | 0.398 | 0.407 | 0.564 | 0.488 | 1.041 | 0.656 | **0.410** | 0.411 | 0.413 | 0.413 | 0.445 | 0.459 | 0.621 | 0.537 | 1.212 0.871 |
| | 720 | 0.469 | 0.441 | 0.674 | 0.538 | 0.670 | 0.532 | 0.667 | 0.541 | 0.454 | 0.440 | 0.592 | 0.504 | 0.950 | 0.636 | **0.410** | 0.450 | 0.474 | 0.453 | 0.543 | 0.490 | 0.671 | 0.561 | 1.166 0.823 |
| | AVG | 0.399 | 0.402 | 0.644 | 0.517 | 0.643 | 0.511 | 0.640 | 0.516 | **0.385** | **0.399** | 0.551 | 0.483 | 0.971 | 0.629 | 0.383 | 0.406 | 0.403 | 0.407 | 0.448 | 0.452 | 0.588 | 0.517 | 0.961 0.734 |
| ETTm2 | 96 | **0.181** | **0.267** | 0.197 | 0.282 | 0.194 | 0.280 | 0.197 | 0.283 | 0.183 | 0.266 | 0.229 | 0.304 | 0.240 | 0.318 | 0.187 | **0.267** | 0.193 | 0.292 | 0.203 | 0.287 | 0.255 | 0.339 | 0.365 0.453 |
| | 192 | **0.247** | **0.308** | 0.260 | 0.320 | 0.258 | 0.318 | 0.261 | 0.321 | 0.251 | 0.310 | 0.287 | 0.338 | 0.301 | 0.352 | **0.249** | 0.309 | 0.284 | 0.362 | 0.269 | 0.328 | 0.281 | 0.340 | 0.533 0.563 |
| | 336 | **0.309** | **0.347** | 0.318 | 0.355 | 0.316 | 0.353 | 0.319 | 0.356 | 0.319 | 0.351 | 0.337 | 0.367 | 0.367 | 0.391 | 0.321 | **0.309** | 0.369 | 0.427 | 0.325 | 0.366 | 0.339 | 0.372 | 1.363 0.887 |
| | 720 | **0.406** | **0.404** | 0.415 | 0.408 | 0.414 | 0.407 | 0.416 | 0.409 | 0.420 | 0.410 | 0.430 | 0.416 | 0.451 | 0.432 | **0.408** | **0.403** | 0.554 | 0.522 | 0.421 | 0.415 | 0.433 | 0.432 | 3.379 1.338 |
| | AVG | **0.286** | **0.332** | 0.297 | 0.341 | 0.295 | 0.340 | 0.298 | 0.342 | 0.293 | 0.334 | 0.321 | 0.356 | 0.340 | 0.373 | **0.291** | **0.322** | 0.350 | 0.401 | 0.305 | 0.349 | 0.327 | 0.371 | 1.410 0.810 |
| Electricity | 96 | 0.198 | **0.282** | 0.375 | 0.469 | 0.391 | 0.478 | 0.310 | 0.406 | 0.196 | 0.287 | 0.232 | 0.321 | 0.198 | 0.290 | 0.168 | 0.272 | 0.197 | **0.282** | **0.193** | 0.308 | 0.201 | 0.317 | 0.274 0.368 |
| | 192 | 0.199 | **0.285** | 0.371 | 0.467 | 0.388 | 0.477 | 0.307 | 0.404 | 0.199 | 0.291 | 0.234 | 0.325 | 0.202 | 0.293 | 0.184 | 0.289 | **0.196** | **0.285** | 0.201 | 0.315 | 0.222 | 0.334 | 0.296 0.386 |
| | 336 | 0.212 | **0.298** | 0.389 | 0.478 | 0.408 | 0.489 | 0.333 | 0.421 | 0.214 | 0.305 | 0.249 | 0.338 | 0.223 | 0.318 | 0.198 | 0.300 | **0.209** | 0.301 | 0.214 | 0.329 | 0.231 | 0.338 | 0.300 0.394 |
| | 720 | 0.253 | **0.330** | 0.424 | 0.497 | 0.447 | 0.511 | 0.384 | 0.450 | 0.254 | 0.335 | 0.289 | 0.366 | 0.259 | 0.341 | 0.220 | 0.320 | **0.245** | 0.333 | 0.246 | 0.355 | 0.254 | 0.361 | 0.373 0.439 |
| | AVG | 0.216 | **0.299** | 0.390 | 0.478 | 0.408 | 0.489 | 0.334 | 0.420 | 0.216 | 0.305 | 0.251 | 0.338 | 0.221 | 0.311 | 0.193 | 0.295 | **0.212** | 0.300 | 0.214 | 0.327 | 0.227 | 0.338 | 0.311 0.397 |
| Weather | 96 | 0.191 | 0.230 | 0.198 | 0.250 | 0.196 | 0.245 | 0.202 | 0.248 | 0.171 | 0.214 | 0.212 | 0.251 | 0.213 | 0.260 | **0.172** | **0.220** | 0.196 | 0.255 | 0.217 | 0.296 | 0.266 | 0.336 | 0.300 0.384 |
| | 192 | 0.236 | 0.267 | 0.250 | 0.290 | 0.248 | 0.286 | 0.254 | 0.288 | 0.217 | 0.254 | 0.261 | 0.288 | 0.269 | 0.300 | **0.219** | **0.261** | 0.237 | 0.296 | 0.276 | 0.336 | 0.307 | 0.367 | 0.598 0.544 |
| | 336 | 0.289 | **0.303** | 0.303 | 0.326 | 0.304 | 0.326 | 0.306 | 0.325 | 0.274 | 0.293 | 0.313 | 0.324 | 0.330 | 0.341 | **0.280** | 0.306 | 0.283 | 0.335 | 0.339 | 0.380 | 0.359 | 0.395 | 0.578 0.523 |
| | 720 | 0.362 | 0.350 | 0.378 | 0.374 | 0.382 | 0.377 | 0.385 | 0.377 | 0.351 | 0.343 | 0.386 | 0.372 | 0.404 | 0.389 | 0.365 | 0.359 | 0.345 | 0.381 | 0.403 | 0.428 | 0.419 | 0.428 | 1.059 0.741 |
| | AVG | 0.270 | 0.288 | 0.282 | 0.310 | 0.282 | 0.308 | 0.287 | 0.309 | 0.253 | 0.276 | 0.293 | 0.309 | 0.304 | 0.323 | **0.259** | **0.287** | 0.265 | 0.317 | 0.309 | 0.360 | 0.338 | 0.382 | 0.634 0.548 |
| Exchange | 96 | 0.081 | 0.201 | 0.102 | 0.225 | 0.100 | 0.221 | 0.098 | 0.218 | **0.086** | **0.209** | 0.142 | 0.261 | 0.137 | 0.260 | 0.107 | 0.234 | 0.088 | 0.218 | 0.148 | 0.278 | 0.197 | 0.323 | 0.847 0.752 |
| | 192 | **0.168** | **0.293** | 0.198 | 0.317 | 0.193 | 0.312 | 0.196 | 0.314 | 0.174 | 0.299 | 0.224 | 0.339 | 0.222 | 0.341 | 0.226 | 0.344 | 0.176 | 0.315 | 0.271 | 0.380 | 0.300 | 0.369 | 1.204 0.895 |
| | 336 | **0.299** | **0.396** | 0.350 | 0.430 | 0.345 | 0.426 | 0.345 | 0.425 | 0.319 | 0.408 | 0.377 | 0.448 | 0.372 | 0.447 | 0.367 | 0.448 | **0.313** | 0.427 | 0.460 | 0.500 | 0.509 | 0.524 | 1.672 1.036 |
| | 720 | **0.805** | **0.674** | 0.905 | 0.721 | 0.889 | 0.715 | 0.883 | 0.712 | 0.875 | 0.701 | 0.939 | 0.736 | 0.912 | 0.727 | 0.964 | 0.746 | **0.839** | **0.695** | 1.195 | 0.841 | 1.447 | 0.941 | 2.478 1.310 |
| | AVG | 0.338 | 0.391 | 0.389 | 0.423 | 0.382 | 0.419 | 0.380 | 0.417 | 0.364 | **0.404** | 0.421 | 0.446 | 0.411 | 0.444 | 0.416 | 0.443 | **0.354** | 0.414 | 0.519 | 0.500 | 0.613 | 0.539 | 1.550 0.998 |
| ILI | 24 | **2.259** | **0.950** | 4.544 | 1.448 | 5.157 | 1.555 | 4.980 | 1.492 | 2.460 | 0.954 | 3.322 | 1.278 | 4.289 | 1.485 | **2.317** | **0.934** | 2.398 | 1.040 | 3.228 | 1.260 | 3.483 | 1.287 | 5.764 1.677 |
| | 36 | 2.239 | 0.936 | 4.619 | 1.493 | 5.620 | 1.692 | 5.593 | 1.658 | **1.998** | **0.912** | 3.696 | 1.374 | 4.360 | 1.510 | 1.972 | 0.920 | 2.646 | 1.088 | 2.679 | 1.080 | 3.103 | 1.148 | 4.755 1.467 |
| | 48 | 1.953 | 0.894 | 4.509 | 1.467 | 5.413 | 1.669 | 5.487 | 1.662 | **1.979** | **0.912** | 3.765 | 1.402 | 4.209 | 1.481 | 2.238 | 0.940 | 2.614 | 1.086 | 2.622 | 1.078 | 2.669 | 1.085 | 4.763 1.469 |
| | 60 | 1.976 | 0.916 | 4.020 | 1.382 | 4.797 | 1.569 | 4.943 | 1.586 | 2.109 | 0.938 | 3.928 | 1.432 | 3.981 | 1.444 | **2.027** | **0.928** | 2.804 | 1.146 | 2.857 | 1.157 | 2.770 | 1.125 | 5.264 1.564 |
| | AVG | 2.107 | 0.924 | 4.423 | 1.448 | 5.247 | 1.621 | 5.251 | 1.600 | **2.137** | **0.929** | 3.678 | 1.372 | 4.210 | 1.480 | 2.139 | 0.931 | 2.616 | 1.090 | 2.847 | 1.144 | 3.006 | 1.161 | 5.137 1.544 |
| Average | | 0.555 | 0.434 | 0.910 | 0.547 | 1.015 | 0.569 | 1.009 | 0.561 | **0.559** | **0.437** | 0.800 | 0.521 | 0.916 | 0.553 | 0.569 | 0.445 | 0.652 | 0.487 | 0.690 | 0.505 | 0.756 | 0.532 | 1.934 0.944 |
| 1st Count | | 32 | | 7 | | 3 | | 0 | | 19 | | 0 | | 0 | | 17 | | 3 | | 4 | | 0 | | 0 |

Table 14: 10% few-shot forecasting results. Yellow : the best in **TY1**; Blue : the second best in **TY1**. Underline: the best over both types; **Bold**: the second best over both types. '-' means 10% time series is not sufficient to constitute a training set.

| Type | | TY1 | | | | | | | | TY2 | | | | | |
|---|---|---|---|---|---|---|---|---|---|---|---|---|---|---|---|
| Method | | TIME-FFM | | FedLoRA | | FedAdapter[H] | | FedAdapter[P] | | UniTime | | GPT4TS | | PatchTST | |
| Metric | | MSE | MAE | MSE | MAE | MSE | MAE | MSE | MAE | MSE | MAE | MSE | MAE | MSE | MAE |
| ETTm1 | 96 | **0.571** | **0.481** | 0.638 | 0.496 | 0.651 | 0.518 | 0.708 | 0.535 | **0.582** | **0.485** | 0.621 | 0.486 | 1.136 | 0.672 |
| | 192 | **0.578** | **0.490** | 0.626 | 0.500 | 0.662 | 0.530 | 0.696 | 0.539 | 0.564 | 0.479 | 0.637 | 0.499 | 1.118 | 0.672 |
| | 336 | **0.592** | **0.504** | 0.628 | 0.506 | 0.666 | 0.540 | 0.686 | 0.543 | 0.578 | 0.489 | 0.648 | 0.508 | 0.987 | 0.637 |
| | 720 | 0.629 | 0.526 | 0.655 | 0.522 | 0.708 | 0.568 | 0.699 | 0.557 | **0.631** | **0.523** | 0.646 | 0.513 | 1.044 | 0.666 |
| | AVG | **0.593** | **0.500** | 0.637 | 0.506 | 0.672 | 0.539 | 0.697 | 0.543 | 0.589 | 0.494 | 0.638 | 0.501 | 1.071 | 0.662 |
| ETTm2 | 96 | **0.195** | **0.277** | 0.198 | 0.282 | 0.200 | 0.284 | 0.201 | 0.287 | 0.192 | 0.274 | 0.197 | 0.278 | 0.255 | 0.329 |
| | 192 | 0.256 | 0.313 | **0.258** | 0.318 | 0.260 | 0.319 | 0.260 | 0.321 | 0.256 | 0.313 | 0.258 | 0.315 | 0.312 | 0.360 |
| | 336 | 0.314 | 0.348 | **0.316** | 0.352 | 0.318 | 0.354 | 0.317 | 0.355 | 0.320 | 0.352 | **0.316** | **0.350** | 0.359 | 0.384 |
| | 720 | **0.412** | **0.403** | 0.415 | 0.407 | 0.415 | 0.407 | 0.413 | 0.407 | 0.429 | 0.413 | 0.410 | 0.402 | 0.465 | 0.440 |
| | AVG | 0.294 | 0.335 | 0.297 | 0.340 | 0.298 | 0.341 | 0.298 | 0.343 | 0.299 | 0.338 | **0.295** | **0.336** | 0.348 | 0.378 |
| Electricity | 96 | 0.249 | 0.329 | 0.253 | 0.341 | 0.404 | 0.478 | 0.391 | 0.474 | **0.236** | 0.327 | 0.231 | 0.316 | 0.344 | 0.416 |
| | 192 | 0.247 | 0.330 | 0.253 | 0.345 | 0.390 | 0.470 | 0.379 | 0.468 | **0.236** | 0.328 | 0.233 | 0.320 | 0.343 | 0.418 |
| | 336 | 0.267 | 0.346 | 0.275 | 0.365 | 0.420 | 0.490 | 0.410 | 0.489 | **0.250** | 0.341 | 0.249 | 0.334 | 0.361 | 0.429 |
| | 720 | 0.300 | **0.368** | 0.319 | 0.400 | 0.469 | 0.518 | 0.452 | 0.513 | **0.295** | 0.371 | 0.292 | 0.365 | 0.399 | 0.453 |
| | AVG | 0.266 | 0.344 | 0.275 | 0.363 | 0.421 | 0.489 | 0.408 | 0.486 | **0.254** | 0.342 | 0.251 | 0.334 | 0.362 | 0.429 |
| Weather | 96 | 0.207 | 0.258 | 0.210 | 0.258 | **0.201** | **0.252** | 0.203 | 0.255 | 0.191 | 0.242 | 0.215 | 0.262 | 0.215 | 0.259 |
| | 192 | 0.259 | 0.297 | 0.265 | 0.301 | **0.254** | **0.293** | 0.255 | 0.295 | 0.240 | 0.278 | 0.270 | 0.304 | 0.265 | 0.297 |
| | 336 | 0.306 | 0.327 | 0.314 | 0.334 | **0.302** | **0.324** | 0.306 | 0.329 | 0.293 | 0.315 | 0.319 | 0.336 | 0.318 | 0.332 |
| | 720 | 0.381 | 0.374 | 0.397 | 0.387 | **0.378** | **0.373** | 0.386 | 0.380 | 0.365 | 0.360 | 0.398 | 0.386 | 0.388 | 0.375 |
| | AVG | 0.288 | 0.314 | 0.296 | 0.320 | **0.284** | **0.311** | 0.287 | 0.315 | 0.272 | 0.299 | 0.300 | 0.322 | 0.297 | 0.316 |
| Exchange | 96 | 0.116 | 0.241 | 0.117 | 0.238 | 0.114 | **0.238** | 0.115 | 0.237 | 0.118 | 0.241 | 0.120 | 0.243 | **0.115** | 0.242 |
| | 192 | 0.212 | 0.331 | 0.218 | 0.333 | 0.209 | 0.329 | 0.211 | 0.329 | 0.208 | 0.328 | 0.221 | 0.337 | 0.197 | 0.321 |
| | 336 | 0.362 | 0.438 | 0.378 | 0.447 | 0.358 | 0.435 | 0.364 | 0.439 | 0.335 | 0.424 | 0.384 | 0.451 | 0.347 | 0.428 |
| | 720 | - | - | - | - | - | - | - | - | - | - | - | - | - | - |
| | AVG | 0.230 | 0.336 | 0.238 | 0.339 | **0.227** | **0.334** | 0.230 | 0.335 | 0.220 | 0.331 | 0.242 | 0.344 | 0.220 | 0.330 |
| Average | | **0.334** | **0.366** | 0.349 | 0.374 | 0.380 | 0.403 | 0.384 | 0.404 | 0.327 | 0.361 | 0.345 | 0.367 | 0.459 | 0.423 |
| 1st Count | | 9 | | 1 | | 1 | | 1 | | 25 | | 12 | | 4 | |

Table 15: Zero-shot forecasting results of Exectricity and Weather with selecting different local parameters. Lower values correspond to better performance. **Bold**: the best.

| Type | ETTh1→ Electricity | | ETTm1→Electricity | | ETTm2→Electricity | | ETTh1→Weather | | ETTm1→Weather | | ETTm2→Weather | |
|---|---|---|---|---|---|---|---|---|---|---|---|---|
| Metric | MSE | MAE | MSE | MAE | MSE | MAE | MSE | MAE | MSE | MAE | MSE | MAE |
| 96 | **0.235** | **0.316** | 0.614 | 0.599 | 0.616 | 0.613 | **0.204** | **0.256** | 0.235 | 0.270 | 0.222 | 0.267 |
| 192 | **0.243** | **0.327** | 0.558 | 0.571 | 0.649 | 0.631 | **0.257** | **0.297** | 0.289 | 0.312 | 0.274 | 0.308 |
| 336 | **0.266** | **0.346** | 0.579 | 0.583 | 0.687 | 0.651 | **0.312** | **0.334** | 0.329 | 0.336 | 0.333 | 0.347 |
| 720 | **0.315** | **0.382** | 0.593 | 0.591 | 0.736 | 0.675 | **0.393** | **0.386** | 0.402 | 0.381 | 0.410 | 0.398 |
| AVG | **0.265** | **0.343** | 0.586 | 0.586 | 0.672 | 0.643 | **0.291** | **0.318** | 0.314 | 0.325 | 0.310 | 0.330 |

Table 16: 5% few-shot forecasting results. Yellow: the best in **TY1**; Blue: the second best in **TY1**. Underline: the best over both types; **Bold**: the second best over both types. '-' means 5% time series is not sufficient to constitute a training set.

| Type | | TY1 | | | | | | | | TY2 | | | | | |
| --- | --- | --- | --- | --- | --- | --- | --- | --- | --- | --- | --- | --- | --- | --- | --- |
| Method | | TIME-FFM | | FedLoRA | | FedAdapter[H] | | FedAdapter[P] | | UniTime | | GPT4TS | | PatchTST | |
| Metric | | MSE | MAE | MSE | MAE | MSE | MAE | MSE | MAE | MSE | MAE | MSE | MAE | MSE | MAE |
| ETTm1 | 96 | 0.515 | 0.459 | 0.557 | 0.462 | 0.585 | 0.492 | 0.585 | 0.489 | 0.576 | 0.498 | 0.591 | 0.499 | 0.559 | 0.477 |
| | 192 | 0.550 | 0.478 | 0.605 | 0.490 | 0.628 | 0.513 | 0.620 | 0.508 | 0.617 | 0.520 | 0.617 | 0.511 | 0.588 | 0.493 |
| | 336 | 0.563 | 0.491 | 0.607 | 0.496 | 0.637 | 0.522 | 0.622 | 0.514 | 0.633 | 0.533 | 0.620 | 0.517 | 0.587 | 0.497 |
| | 720 | 0.641 | 0.536 | 0.655 | 0.529 | 0.750 | 0.579 | 0.715 | 0.566 | 1.028 | 0.680 | 0.694 | 0.561 | 0.631 | 0.522 |
| | AVG | 0.567 | 0.491 | 0.606 | 0.494 | 0.650 | 0.526 | 0.636 | 0.519 | 0.713 | 0.558 | 0.631 | 0.522 | 0.591 | 0.497 |
| ETTm2 | 96 | 0.192 | 0.272 | 0.196 | 0.278 | 0.196 | 0.278 | 0.194 | 0.277 | 0.198 | 0.279 | 0.198 | 0.282 | 0.200 | 0.282 |
| | 192 | 0.254 | 0.311 | 0.260 | 0.318 | 0.259 | 0.317 | 0.258 | 0.316 | 0.266 | 0.323 | 0.259 | 0.317 | 0.260 | 0.318 |
| | 336 | 0.312 | 0.346 | 0.318 | 0.352 | 0.318 | 0.352 | 0.316 | 0.351 | 0.337 | 0.366 | 0.316 | 0.351 | 0.318 | 0.352 |
| | 720 | 0.415 | 0.403 | 0.419 | 0.408 | 0.420 | 0.410 | 0.418 | 0.408 | 0.453 | 0.430 | 0.417 | 0.407 | 0.419 | 0.407 |
| | AVG | 0.293 | 0.333 | 0.298 | 0.339 | 0.298 | 0.339 | 0.296 | 0.338 | 0.313 | 0.350 | 0.298 | 0.339 | 0.299 | 0.339 |
| Electricity | 96 | 0.312 | 0.394 | 0.326 | 0.407 | 0.318 | 0.398 | 0.320 | 0.397 | 0.281 | 0.371 | 0.256 | 0.339 | 0.295 | 0.379 |
| | 192 | 0.305 | 0.391 | 0.327 | 0.414 | 0.312 | 0.398 | 0.313 | 0.396 | 0.283 | 0.377 | 0.254 | 0.341 | 0.293 | 0.382 |
| | 336 | 0.321 | 0.401 | 0.340 | 0.422 | 0.338 | 0.417 | 0.335 | 0.412 | 0.294 | 0.385 | 0.271 | 0.354 | 0.308 | 0.392 |
| | 720 | 0.358 | 0.427 | 0.365 | 0.436 | 0.364 | 0.433 | 0.365 | 0.430 | 0.335 | 0.413 | 0.313 | 0.385 | 0.341 | 0.413 |
| | AVG | 0.324 | 0.403 | 0.339 | 0.420 | 0.333 | 0.411 | 0.333 | 0.409 | 0.298 | 0.387 | 0.273 | 0.355 | 0.309 | 0.391 |
| Weather | 96 | 0.214 | 0.265 | 0.222 | 0.269 | 0.212 | 0.262 | 0.219 | 0.267 | 0.209 | 0.260 | 0.207 | 0.259 | 0.221 | 0.271 |
| | 192 | 0.264 | 0.302 | 0.275 | 0.310 | 0.263 | 0.301 | 0.270 | 0.305 | 0.258 | 0.297 | 0.258 | 0.297 | 0.271 | 0.308 |
| | 336 | 0.310 | 0.329 | 0.321 | 0.338 | 0.311 | 0.330 | 0.319 | 0.335 | 0.306 | 0.325 | 0.308 | 0.328 | 0.318 | 0.336 |
| | 720 | 0.381 | 0.374 | 0.394 | 0.385 | 0.383 | 0.376 | 0.393 | 0.382 | 0.380 | 0.371 | 0.380 | 0.373 | 0.391 | 0.382 |
| | AVG | 0.292 | 0.317 | 0.303 | 0.325 | 0.292 | 0.317 | 0.300 | 0.322 | 0.288 | 0.313 | 0.288 | 0.314 | 0.301 | 0.324 |
| Exchange | 96 | 0.118 | 0.244 | 0.121 | 0.244 | 0.117 | 0.243 | 0.116 | 0.241 | 0.385 | 0.458 | 0.120 | 0.246 | 0.123 | 0.250 |
| | 192 | 0.215 | 0.334 | 0.221 | 0.337 | 0.215 | 0.333 | 0.215 | 0.333 | 0.498 | 0.528 | 0.216 | 0.334 | 0.220 | 0.337 |
| | 336 | - | - | - | - | - | - | - | - | - | - | - | - | - | - |
| | 720 | - | - | - | - | - | - | - | - | - | - | - | - | - | - |
| | AVG | 0.167 | 0.289 | 0.171 | 0.291 | 0.166 | 0.288 | 0.166 | 0.287 | 0.442 | 0.493 | 0.168 | 0.290 | 0.171 | 0.293 |
| Average | | 0.329 | 0.367 | 0.344 | 0.374 | 0.348 | 0.376 | 0.346 | 0.375 | 0.411 | 0.420 | 0.332 | 0.364 | 0.334 | 0.369 |
| 1st Count | | 20 | | 0 | | 3 | | 6 | | 8 | | 17 | | 2 | |

Table 17: Zero-shot forecasting results. Lower values correspond to better performance. Yellow: the best in **TY1**; Blue: the second best in **TY1**. Underline: the best over both types; **Bold**: the second best over both types.

| Type | | TY1 | | | | | | | | TY2 | | | | | |
| --- | --- | --- | --- | --- | --- | --- | --- | --- | --- | --- | --- | --- | --- | --- | --- |
| Method | | TIME-FFM | | FedIT | | FedAdapter[H] | | FedAdapter[P] | | UniTime | | GPT4TS | | PatchTST | |
| Metric | | MSE | MAE | MSE | MAE | MSE | MAE | MSE | MAE | MSE | MAE | MSE | MAE | MSE | MAE |
| ETTh2 | 96 | 0.296 | 0.344 | 0.303 | 0.351 | 0.303 | 0.351 | 0.304 | 0.352 | 0.306 | 0.352 | 0.316 | 0.361 | 0.332 | 0.371 |
| | 192 | 0.373 | 0.391 | 0.391 | 0.401 | 0.391 | 0.402 | 0.390 | 0.401 | 0.389 | 0.401 | 0.400 | 0.410 | 0.422 | 0.421 |
| | 336 | 0.410 | 0.424 | 0.425 | 0.432 | 0.426 | 0.434 | 0.425 | 0.432 | 0.424 | 0.434 | 0.430 | 0.439 | 0.462 | 0.455 |
| | 720 | 0.413 | 0.437 | 0.428 | 0.443 | 0.431 | 0.447 | 0.428 | 0.444 | 0.433 | 0.450 | 0.442 | 0.461 | 0.467 | 0.469 |
| | AVG | 0.373 | 0.399 | 0.387 | 0.407 | 0.388 | 0.408 | 0.387 | 0.407 | 0.388 | 0.409 | 0.397 | 0.418 | 0.421 | 0.429 |
| Electricity | 96 | 0.235 | 0.316 | 0.392 | 0.464 | 0.383 | 0.460 | 0.395 | 0.470 | 0.409 | 0.481 | 0.448 | 0.520 | 0.529 | 0.562 |
| | 192 | 0.243 | 0.327 | 0.376 | 0.455 | 0.376 | 0.458 | 0.384 | 0.466 | 0.410 | 0.484 | 0.443 | 0.517 | 0.507 | 0.550 |
| | 336 | 0.266 | 0.346 | 0.397 | 0.471 | 0.404 | 0.477 | 0.412 | 0.484 | 0.439 | 0.504 | 0.462 | 0.526 | 0.536 | 0.566 |
| | 720 | 0.315 | 0.382 | 0.428 | 0.490 | 0.441 | 0.499 | 0.446 | 0.506 | 0.487 | 0.531 | 0.494 | 0.542 | 0.563 | 0.581 |
| | AVG | 0.265 | 0.343 | 0.398 | 0.470 | 0.401 | 0.474 | 0.409 | 0.482 | 0.436 | 0.500 | 0.462 | 0.526 | 0.534 | 0.565 |
| Weather | 96 | 0.204 | 0.256 | 0.212 | 0.261 | 0.220 | 0.266 | 0.218 | 0.265 | 0.210 | 0.262 | 0.223 | 0.271 | 0.235 | 0.277 |
| | 192 | 0.257 | 0.297 | 0.266 | 0.302 | 0.272 | 0.306 | 0.271 | 0.306 | 0.264 | 0.303 | 0.287 | 0.319 | 0.293 | 0.320 |
| | 336 | 0.312 | 0.334 | 0.314 | 0.334 | 0.319 | 0.337 | 0.320 | 0.338 | 0.326 | 0.334 | 0.347 | 0.357 | 0.351 | 0.356 |
| | 720 | 0.393 | 0.386 | 0.389 | 0.381 | 0.397 | 0.387 | 0.398 | 0.388 | 0.402 | 0.382 | 0.432 | 0.409 | 0.427 | 0.404 |
| | AVG | 0.291 | 0.318 | 0.295 | 0.319 | 0.302 | 0.324 | 0.302 | 0.324 | 0.301 | 0.320 | 0.322 | 0.339 | 0.327 | 0.339 |
| Average | | 0.310 | 0.353 | 0.360 | 0.399 | 0.364 | 0.402 | 0.366 | 0.404 | 0.375 | 0.410 | 0.394 | 0.428 | 0.427 | 0.444 |

Table 18: Mean values and standard deviations of **TY1**.

| Method | TIME-FFM | | FedLoRA | | FedAdapter[H] | | FedAdapter[P] | |
|---|---|---|---|---|---|---|---|---|
| Metric | MSE | MAE | MSE | MAE | MSE | MAE | MSE | MAE |
| ETTh1 | **0.446±0.006** | **0.434±0.001** | 0.483±0.002 | 0.462±0.001 | 0.478±0.020 | 0.467±0.013 | 0.496±0.008 | 0.482±0.004 |
| ETTh2 | 0.383±0.001 | 0.407±0.001 | **0.375±0.001** | **0.397±0.002** | 0.375±0.002 | 0.399±0.001 | 0.377±0.003 | 0.401±0.002 |
| ETTm1 | **0.398±0.001** | **0.402±0.001** | 0.667±0.020 | 0.523±0.006 | 0.641±0.041 | 0.516±0.013 | 0.673±0.029 | 0.529±0.012 |
| ETTm2 | **0.286±0.001** | **0.331±0.000** | 0.298±0.001 | 0.343±0.001 | 0.298±0.003 | 0.343±0.003 | 0.299±0.001 | 0.344±0.002 |
| Electricity | **0.216±0.002** | **0.299±0.002** | 0.377±0.012 | 0.464±0.012 | 0.359±0.059 | 0.449±0.050 | 0.368±0.030 | 0.457±0.032 |
| Weather | **0.274±0.005** | **0.291±0.004** | 0.284±0.002 | 0.310±0.001 | 0.283±0.002 | 0.310±0.003 | 0.285±0.002 | 0.311±0.002 |
| Exchange | **0.349±0.017** | **0.396±0.008** | 0.389±0.002 | 0.423±0.001 | 0.384±0.002 | 0.421±0.002 | 0.380±0.001 | 0.418±0.001 |
| ILI | **2.250±0.146** | **0.969±0.048** | 4.712±0.250 | 1.510±0.054 | 4.557±0.621 | 1.516±0.093 | 4.658±0.518 | 1.517±0.072 |

