# OpenReview forum: "Time-FFM: Towards LM-Empowered Federated Foundation Model for Time Series Forecasting"
_NeurIPS.cc/2024/Conference — NeurIPS 2024 poster_

### Official Review · Reviewer_7qPS · 2024-06-14

**Soundness:** 3
**Presentation:** 4
**Contribution:** 3
**Rating:** 7
**Confidence:** 5

**Summary:**

The authors study an important problem of time series forecasting considering the increasing concerns of privacy and copyright. The paper proposes a novel LLM-empowered federated time series forecasting model with three main components, i.e., modality alignment, prompt adaption, and personalized strategy.

**Strengths:**

1. The paper is well-written and easy to follow.
2. The authors propose Time-FFM, a novel LLM-empowered federated foundation model for time series forecasting. Time-LLM encompasses a cross-modality module, a prompt adaption module, and a personalized federated training strategy.
3. Extensive experiments show the effectiveness of the proposed method.

**Weaknesses:**

1. We often run federated learning on edge devices, which have restrictedly limited data processing capability. Time-FFM is difficult to deploy in real-world scenarios. Additionally, the high latency of LLM during inference may impede its applicability for real-time time series forecasting. This presents a practical challenge for deployment in resource-limited FL environments. Moreover, it requires at least a GPU to run LLM which is impractical and unrealistic in real-world FL scenarios.
2. TimesFM [r1], a centralized foundation model for time series forecasting from Google, was accepted by ICML 2024. It was released on Arxiv last October. Google possesses various kinds of data from a variety of domains without any copyright and privacy concerns, which conflicts with your motivation.

    [r1]. Das, et al., A decoder-only foundation model for time-series forecasting, ICML, 2024.

3. More detailed explanations are needed regarding how the proposed method addresses data heterogeneity across clients.
4. It would be better to improve and clarify the technique depth. Patching and channel-independent mechanisms are widely used in existing time series forecasting methods. In addition, the prompt adaption seems to be another version of an existing technique [22]. The projection head is also simple and straightforward. Further, the training process of Time-LLM is based on FedAvg.
5. It is suggested to illustrate more on personalized strategy including the global encoder. In addition, it is also confusing how to deal with the conflicts between generalization and personalization.
6. It would be better to convert TimesNet, DLiner, and PatchTST, into their federated version with FedAvg and compare their federated versions with the proposed Time-FFM. Additionally, it would be better to include a case study to intuitively show how accurately the proposed Time-FFM can predict time series.

**Questions:**

1. Are the practical deployment challenges considered?
2. How to solve the bottleneck problem of limited FL resources? Could you provide more evidence?
3. How to demonstrate the rationality of the prompt generated by LLM? How to justify the generated prompt is related to the corresponding domain? It would be better to provide more case studies on all datasets for a more comprehensive evaluation. In addition, it is especially encouraged to provide a corresponding theoretical analysis.

**Limitations:**

The main limitation is the practicality of the proposed Time-FFM. In addition, please see the weaknesses and questions for other limitations.

---

> ### Author Rebuttal · Authors · 2024-08-05
>
> We show great gratitude to you for approving of the quality of our paper. We have make the detailed response to thoroughly release your concerns.
>
> > W1 & Q1 & Q2: Concerns on the applicability in real world.
>
> We are sorry for not covering the deployment of Time-FFM in the manuscript. Since we make Time-FFM a "domain" level time series forecaster, we deem that federated training participants are more likely to be **edge clouds with abundant computing resources**, instead of resource-limited edge devices. Upon training, Time-FFM can also be deployed at edge clouds for forecasting tasks. Hence, we think that the resource limitation might not be a bottleneck of the deployment of Time-FFM and it realistic to deploy Time-FFM in real-world FL scenarios.
>
> > W2: TimesFM, as a centralized foundation model, conflicts with your motivation.
>
> Compared with the modality of text, time series is more domain-specific and (commercial) copyright-sevsitive, i.e., private knowledge may be inferred from historical time series readings, especilly in finance and healthcare domain. Hence, it is of great significance to take data privacy into consideration in the construction of time series foundation models. Moreover, a multitude of public data cannot even be adopted for pre-training foundation models due to data license restriction. Some foundation models like MOIRAI, have released large-scale public time series data merely for research instead of commercial use.
>
> Hence, our work uniquely **bridges the gap between foundation models and federated learning**, which not only enhances the privacy and applicability of foundation models in sensitive domains but also opens up new avenues for leveraging rich, yet previously inaccessible, time series data for advanced predictive analytics, addressing a crucial need in the (commercial) field.
>
> > W3&W5: More details for how to tackle data heterogeneity across clients.
>
> We aim to learn the underlying commonalities present in time sereis data, though corss-domain time series data are of great heterogeneity. In Federated Averaging framework, all domains learn a single shared model which would **perform poorly on each specific domain**. If we simply train dedicated model for each domain using their own local data (even if it is small) to train a local model, the domain-specific models may not generalize well to unseen domains. Therefore, we need to **strike a balance between the generalization and domain-specific prediction**. We devise a persoanlized federated strategy with a global encoder and personalized heads. The global shared encoder can learn the** common temporal embeddings or representations across domains**, which is in line with the success of centralized deep learning, i.e., the sharing of global feature representations among data [1][2]. The personalized heads enables **domain-specific decoders**, accounting for yielding prediction results fitting for local distributions.
>
> [1] LeCun, Y., Bengio, Y., and Hinton, G. Deep learning. nature, 2015.
> [2] Bengio, Y., Courville, A., and Vincent, P. Representation learning: A review and new perspectives. IEEE TPAMI, 2013.
>
> > W4: It would be better to improve and clarify the technique depth.
>
> Thanks for your insightful concerns on our contributions. We aim to propose a Language model-empowered federated foundation model for time series forecasting.
>
> - Given the differentiation of dimensionality and horizon, we introduce the modality alignment module encomposing the channel-independent and patching techniques, which may follow the track of GPT4TS, Time-LLM, MOIRAI, etc.
> - For bootstrapping the pre-trained GPT2 backbone for cross-domain time series reasoning, we propose to **adaptively construct prompts from how to understand patch tokens**, rather than from rigid domain instructions like Time-LLM and UniTime.
> - Due to cross-domain time series heterogeneity, we devise a personalized federated strategy (different from Federated Averaging which aims at learning a global prediction model), with **global encoder and personalized prediction heads**.
>
> In conclusion, we propose the first federated foundation model for time series forecasting, with adaptively generating domain-specific prompts and tackling time series heterogeneity for general-purpose learning and personzalied prediction.
>
> > W6: It would be better to convert TimesNet, DLiner, and PatchTST, into their federated version.
>
> Thanks for your valuable suggeations. We report the performance comparison in **Table 3 of the PDF in the "Author Rebuttal"**. In the modified version, we will supplement the experiment results and the prediction case study to make the effectiveness of Time-FFM more convincing.
>
> > Q3: The rationality of the generated prompts.
>
> We appreciate your attention on the rationality of prompts. We want to adaptively select the "optimal" text prototypes as prompts according to the domain-sepecific patch tokens by prompt adaption module. We think the prompts indicate how the LM backbone understand the domain temporal pattterns (conveyed by patch tokens) and thus can bootstrap domain-sepecific time series reasoning, though the prompts may have conflict with how we humans understand the patch segment [1].
>
> We evaluate the effectiveness of prompts in Table 5. We have the key observation that the prompts can improve the forecasting performance (by comparing A.1 and A.2); the proposed adaptive prompt outperforms the domain instruction (by comparing A.1 and A.3). Furthermore, the showcase in Fig. 3 illustrates that different domain datasets correspond different text prototypes (say prompts).
>
> Hence, we think ablation results and showcase can both reflect the rationality of prompt adaption. In the modified version, we will follow your suggestion to provide more case studies and theoretical analysis to make it more convincing.
>
> [1] Sun, C., Li, H., Li, Y., & Hong, S. TEST: Text Prototype Aligned Embedding to Activate LLM's Ability for Time Series. In ICLR 2024.

---

> > ### Comment · Reviewer_7qPS · 2024-08-08
> >
> > Thanks for the response, which addresses most problems. However, there are still some minor points to be clarified.
> > 1. If we consider the clients as edge clouds, which are often possessed by big companies, such as Google, many kinds of time series data are supposed to be included in these edge clouds. In such situations, training an LLM-based FL method would be less important, considering that data access restriction is not a big problem.
> > 2. It would be better to explain the personalized federated strategy in detail for better understanding.
> > 3. It is suggested to improve the draft by adding a section (or contents) to clarify the technical depth according to the response to W4.

---

> ### Author Response · Authors · 2024-08-09
> **Response to Reviewer 7qPS**
>
> We are happy that our responses have effectively addressed your concerns. We would like to express our gratitude again for taking the time to review our paper and provide us with such detailed and invaluable suggestions.
>
> >C1. If we consider the clients as edge clouds, which are often possessed by big companies, such as Google, many kinds of time series data are supposed to be included in these edge clouds. In such situations, training an LLM-based FL method would be less important, considering that data access restriction is not a big problem.
>
> Thanks for providing such insightful point. We will make analysis from the perspectives of the necessity of federated learning and the adoption of the LLM-based backbone.
>
> **Necessity of Federated Learning.**  Actually, in the big company on the scale of Google, there exists data restriction across different departments. The time series data, especially from different bussiness scenarios, can hardly be transmitted directly to a centralized cloud server for training, given the **data volume** and **access limitation**. Hence, a foundation model trained in FL paradigm may be an optimal choice. On the other hand, different (big or small) companies can be united in the community for construct a in- or cross-domain foundation model.
>
> **Adoption of LLM.** We adopt the first 6 transformer layers of GPT2 as the backbone. This also follows the track of training-from-scratch foundation models, such as MOIRAI and TimesFM, where multiple transformer layers are also adopted as the backbone. In our paper, we merely initialize the backbone with the pre-trained GPT2 parameters, which can achieve comparable performance in full-tuning or fine-tuning. As a matter of fact, with the available large-scale time series data, we can train our Time-FFM from scratch without being initilized by GPT2 parameters, like Chronos.
>
> >C2. It would be better to explain the personalized federated strategy in detail for better understanding.
>
> Thanks for your valuable suggestion. **Firstly, we want to clarify that domain personalization is a necessity.** The foundation model can learn the underlying commonalities in time series data, but may not guarantee the prediction performance for a specific domain, if we directly adopt the Federated Averaging framework to learn shared global model parameters for each domain. While in centralized training mode, the pre-trained model needs to be fine-tuned firstly and then adopted for specific forecasting tasks to ensure the (personalized) prediction performance.
>
> **Secondly, we analyze the rationality and present the techinical details of the designed personalized strategy.** We are inspired by the success of the centralized learning, where heterogenous data are projected to representations of the shared latent space. Hence, we can learn a global encoder to enable a shared representation space over different domains and dedicated local prediction heads for generating personalized prediction results in line with local distribution.
>
> We will modify the manuscript according to the response to W3 and C2, and supplement a case study to illustrate the rationality of personalized strategy. We sincerely hope such revision is under your consideration.
>
> >C3. It is suggested to improve the draft by adding a section (or contents) to clarify the technical depth according to the response to W4.
>
> We will carefully follow your suggestion to clarify the technical depth in the revised version to making the novelty of the proposed Time-FFM more convincing.

---

> > ### Comment · Reviewer_7qPS · 2024-08-09
> >
> > Thanks for the response, which makes sense and addresses all my questions.

---

> > > ### Author Response · Authors · 2024-08-10
> > > **Gratitude to Reviewer 7qPS**
> > >
> > > We are happy that our responses have fully released your concerns. We would like to thank you for your professional review work, constructive comments, and valuable suggestions on our manuscript.

---

### Official Review · Reviewer_uvxC · 2024-07-12

**Soundness:** 3
**Presentation:** 3
**Contribution:** 2
**Rating:** 7
**Confidence:** 5

**Summary:**

This paper proposed a federated foundation model for time series forecasting. This foundation model is trained in a distributed setting, with global shared parameters in a server, and domain / dataset specific parameters in clients. This allows the authors to personalize their predictions to local domain-specific data, while still learning general time series patterns. The global parameters belong to the prompt adaptation and modality alignment modules, whereas the local parameters include word embeddings, frozen LLMs, and client specific prediction heads. These methods are then compared with federated fine-tuning, other LLM adaptation methods for time series forecasting, and specialized time series models on standard time series forecasting datasets.

**Strengths:**

1. The paper is well written and easy to understand.
2. The authors conduct multiple experiments including ablation experiments, which are well executed.

**Weaknesses:**

1. **Missing literature:** The authors do not cite or discuss two bodies of work which are pertinent to this discussion: (1) literature on time series foundation models, including models such as LagLLama [1], MOIRAI [2], TimesFM [3], Moment [4], Chronos [5] etc., (2) recent time series forecasting methods such as iTransformer [6], or older techniques such as N-BEATs [7] and N-HITS [8] etc. which have been shown to be strong time series forecasters, and (3) literature on federated learning for time series data.
2. **Rationale and contributions:** I am not sure about the motivation behind the work. I feel that foundation modeling and federated learning seem at odds with each other, at least in this case. The authors argue that data owners may not be willing to share their data. But most time series foundation models, or even LLMs and LLVMs are trained on public data, and later adapted to private data, so I don't see the value proposition of such an approach.  Also I am not sure how this is a foundation model, as it doesn't solve multiple tasks, and is neither trained on multiple large-scale datasets. Secondly, why use large language models for time series forecasting, when studies have shown that there is sufficient data to pre-train foundation models from scratch [4], and that LLMs may not be better at forecasting time series data despite their significant computational cost [9]. Finally, I am not sure what the contributions of the work are. It seems like most of the work is built on architectural design choices from PatchTST, TimeLLM and Federated Averaging. Also, the personalization aspect sounds interesting, but can be addressed by fine-tuning foundation models, or custom built forecasting methods trained on specific datasets.
3. **Baselines and comparisons:** The authors do not compare their methods with any recent time series foundation model, or recent time series forecasting models such as iTransformer, and old but performant methods such as N-BEATs / N-HITS. Recent studies such as [] have shown that older methods get the better of most recent methods in many settings.

### References
1. Rasul, Kashif, et al. "Lag-llama: Towards foundation models for time series forecasting." arXiv preprint arXiv:2310.08278 (2023).
2. Woo, Gerald, et al. "Unified training of universal time series forecasting transformers." arXiv preprint arXiv:2402.02592 (2024).
3. Das, Abhimanyu, et al. "A decoder-only foundation model for time-series forecasting." arXiv preprint arXiv:2310.10688 (2023).
4. Goswami, Mononito, et al. "Moment: A family of open time-series foundation models." arXiv preprint arXiv:2402.03885 (2024).
5. Ansari, Abdul Fatir, et al. "Chronos: Learning the language of time series." arXiv preprint arXiv:2403.07815 (2024).
6. Liu, Yong, et al. "itransformer: Inverted transformers are effective for time series forecasting." arXiv preprint arXiv:2310.06625 (2023).
7. Oreshkin, Boris N., et al. "N-BEATS: Neural basis expansion analysis for interpretable time series forecasting." arXiv preprint arXiv:1905.10437 (2019).
8. Challu, Cristian, et al. "Nhits: Neural hierarchical interpolation for time series forecasting." Proceedings of the AAAI conference on artificial intelligence. Vol. 37. No. 6. 2023.
9. Tan, Mingtian, et al. "Are Language Models Actually Useful for Time Series Forecasting?." arXiv preprint arXiv:2406.16964 (2024).

**Questions:**

I do not have any specific questions from the authors. But I would appreciate some clarity on the motivation and rationale behind the method. Please see my arguments above.

**Limitations:**

The authors have discussed some limitations of their work. I believe that the authors should mention the computational cost of their method as a limitation. Each dataset / domain / client has a large language model backbone + other parameters, which makes their method computationally expensive.

----

After rebuttal: discussions on legality of using non-permissively licensed datasets to train FMs and notions of privacy need to be discussed in the paper.

---

> ### Author Rebuttal · Authors · 2024-08-05
>
> We are deeply grateful for the insightful review you have provided for our manuscript. We have made the following response.
>
> > W1 & W3: Missing literature on **(1)** time series foundation models, **(2)** deep learning methods, and **(3)** federated learning methods.
>
> We show great appreciation that the suggested literature is of great value to improve our manuscript. We will cite these works and report the corresponding performance comparison in the revision.
>
> **For (1):** Works in **(1)** aim to construct a time series "corpus" and then train a time series foundation model from scratch. The performance comparison is reported in **Table 1 of the attachment PDF in the "Author Rebuttal"**.
>
> **For (2):** We compare multiple time series forecasting methods in the manuscript and feel sorry for not covering the relevant methods you metioned. We report the performance comparison in **Table 2 of the attachment PDF in the "Author Rebuttal"**.
>
> **For (3):** There have not any literature focusing on building foundation models for time series forecasting. Therefore, we merely compare the federated fine-tuning methods in **TY1**.
>
> > W2.1: Rationale and contributions: **(1)** The motivation behind the work. **(2)** How this is a foundation model.
>
> **For (1):** We agree with you that LLMs or LLVMs are trained on public data. Compared with the modality of text, time series is more domain-specific and (commercial) copyright-sevsitive, i.e., **private knowledge may be inferred from historical time series readings**, especilly in finance and healthcare domain. Hence, it is of great significance to take data privacy into account when constructing time series foundation models. Moreover, a multitude of public data cannot even be adopted for pre-training foundation models due to data license restriction, such as Kaggle public datasets. Some foundation models like MOIRAI, have released large-scale public time series data **merely for research instead of commercial use**.
>
> Hence, our work uniquely **bridges the gap between foundation models and federated learning**, which not only enhances the privacy and applicability of foundation models in sensitive domains but also opens up new avenues for leveraging rich, yet previously inaccessible, time series data for advanced predictive analytics, addressing a crucial need in the (commercial) field.
>
> **For (2):** We design Time-FFM as a foundation model for **time series forecasting task**, like MOIRAI and Lag-Llama. Due to the absence of public time series repositories, we merely train Time-FFM on 8 benchmark datasets. **Actually, Time-FFM can be trained on the built large-scale time series archive in MOIRAI or Lag-LIama** and then applied for downstream forecasting tasks.
>
> > W2.2: Why use LLMs.
>
> In our manuscript, we adopt the first 6 Transformer layers of the pre-trained GPT2 in Time-FFM, while in the foundation models training from scratch (MOIRAI, Lag-Llama, MOMENT and TimesFM), stacked Transformer layers are also adopted as the encoder backbone. We simply freeze all parameters of Transformer layers, which can achieve comparable performance compared to fine-tuning or full-tuning (observed from Table 6 in the manuscript). Actually, with the available large-scale time series data, we can train our Time-FFM from scratch without being initilized by GPT2 parameters, like Chronos.
>
> > W2.3: **(1)** The contributions of the work and **(2)** necessity of personalized heads.
>
> **For (1)**: We aim to propose a LM-empowered federated foundation model for time series forecasting.
>
> - Given the differentiation of dimensionality and horizon, we introduce the modality alignment module encomposing the channel-independent and patching techniques, which may follow the track of GPT4TS, Time-LLM, MOIRAI, Moment, etc.
> - For bootstrapping the pre-trained GPT2 backbone for cross-domain time series reasoning, we propose to **adaptively construct prompts from how to understand patch tokens**, rather than **from rigid domain instructions like Time-LLM and UniTime**.
> - Due to cross-domain time series heterogeneity, we devise a personalized federated strategy (different from Federated Averaging which aims at learning a global prediction model), with **global encoder and personalized prediction heads**.
>
> In conclusion, we propose the first federated foundation model for time series forecasting, with adaptively generating domain-specific prompts and tackling time series heterogeneity for general-purpose learning and personzalied prediction.
>
> **For (2)**: Our proposed persoanlized strategy is coherent to the federated training process. Each domain participant merely uploads the updated parameters of encoders, **without additional modification of local optimization process**. Upon finishing federated training, a global encoder and multiple heads (one for each domain) are obtained. When applying to the downstream forecasting tasks, the global encoder and in-domain heads can be frozen and directly adopted, avoiding the process of fine-tuning.
>
> > L1: Limitations on computational cost.
>
>  We agree with you that the computational cost is potentially high even we freeze all parameters of the LM backbone. In the revised version, we will follow your suggestion by supplementing such limitation.
>
> Furthermore, since Time-FFM works as a "domain" level time series forecaster, we think that federated training participants are more likely to be edge clouds with abundant computing resources, instead of resource-limited terminal devices. Upon training, Time-FFM can also be deployed at edge clouds for forecasting tasks. Hence, we think that the resource limitation might not be a bottleneck of the deployment of Time-FFM in real world.
>
> We will carefully incorporate your comments in the revised paper. Considering the encouraging comments from Reviewers LqRs, PDR4, and 7qPS,  we believe our research findings are worth sharing with the research community. We sincerely hope that a revision is still considered.

---

> > ### Author Response · Authors · 2024-08-11
> > **Kindly Request for Reviewer's Feedback**
> >
> > Dear Reviewer uvxC,
> >
> > Since the End of author/reviewer discussions is coming soon, may we know if our response addresses your main concerns? If so, we kindly ask for your reconsideration of the score. Should you have any further advice on the paper and/or our rebuttal, please let us know and we will be more than happy to engage in more discussion and paper improvements.
> >
> > Thank you so much for devoting time to improving our paper!

---

> > > ### Comment · Reviewer_uvxC · 2024-08-12
> > > **Thank you for the rebuttal!**
> > >
> > > Dear Authors,
> > >
> > > I really appreciate the time and effort that you have put into the rebuttal! I am inclined to increase my score if the authors can provide answers to the following core concerns that remain:
> > >
> > > 1. The authors claim that "*private knowledge may be inferred from historical time series readings.*" Could the authors provide some evidence for this statement. Intuitively, I would expect text data to contain more personal identifiable information (name, address, age, gender, etc.) in comparison to time series data. I am not an expert in privacy, so any clarity on this matter would greatly improve my perception of the motivation of this work.
> > > 2. Data licensing is an important and interesting issue. I do think there is sufficient public time series dataset available with permissive licensing to train a "foundation model" from scratch or adapt a pre-training model.
> > > 3. "*Each domain participant merely uploads the updated parameters of encoders, without additional modification of local optimization process.*" I do not understand this sentence. How are the parameters of the encoders updates without local optimization process? Or am I not understanding what you mean?
> > > 4. I still think that the authors are using federated averaging. Parameters being shared by clients and averaged in a server is exactly what federated averaging does. Is there anything I am missing?
> > > 5. The authors misunderstood "(3) federated learning methods". I meant the authors should compare and contrast their work with literature on time series models and federated learning. A [Google scholar search](https://scholar.google.com/scholar?hl=en&as_sdt=0%2C33&q=federated+forecasting+time+series+&btnG=) yields some seemingly relevant work.
> > >
> > > Again, in there interest of time, I do not expect any new experiments. However, I would appreciate some discussion on these points based on existing work.
> > >
> > > Thanks!

---

> > > > ### Author Response · Authors · 2024-08-13
> > > > **Response to Reviewer uvxC**
> > > >
> > > > We would like to express our gratitude for taking the time to review our paper. We really appreciate your increased score and will carefully respond to your comments as follows. We hope such response can release your concerns.
> > > >
> > > > >C1. Clarity on privacy leakage in time series data.
> > > >
> > > > Thanks for your insightful comment. For example, the healthcare time series data contain the historical observations on various indicators of patients, such as blood pressure, blood glucose, etc, which the patients are reluctant to share with the third party even after anonymization [1][2]. Furthermore, in commercial cooperation scenario, each company may not share the access to its own data, avoiding leaking private assets [3][4]. Hence, we think it is of great significance to **keep private time series data locally, especially for the sensitive domains**. Therefore, we think Time-FFM bridges the gap between foundation models and federated learning, which opens up new avenues for leveraging rich, yet previously inaccessible, time series data for advanced predictive analytics.
> > > >
> > > > [1] Lian, Wang, et al. Blockchain-based personalized federated learning for internet of medical things. IEEE Transactions on Sustainable Computing, 2023.
> > > >
> > > > [2] Rauniyar, Hagos, et al. Federated learning for medical applications: A taxonomy, current trends, challenges, and future research directions. IEEE IoTJ, 2023.
> > > >
> > > > [3] Liu, James, et al. Privacy-preserving traffic flow prediction: A federated learning approach. IEEE IoTJ, 2020.
> > > >
> > > > [4] Zhang, Zhang, et al. FASTGNN: A topological information protected federated learning approach for traffic speed forecasting. IEEE TII, 2021.
> > > >
> > > > >C2. There is sufficient public time series dataset for training foundation models.
> > > >
> > > > The time series foundation models such as Moirai and TimesFM are indeed pre-trained based on the available time series data. However, these foundation models are merely oriented for **research rather than commercial use**. Furthermore, in real-world application, we aim to train a foundation model, which can perform well on all or most downstream tasks of a certain domain, such as the domain of transportation, finance, etc. While the available public data may be not domain-concentrated so that the generated foundation model could not **"learn the domain-specific knowledge"**. Given such data sparsity, a better way is to combine various datasets of the domain which may be privately held by different parties, to train a foundation model in federated learning paradigm.
> > > >
> > > > >C3. How are the parameters of the encoders updates without local optimization process?
> > > >
> > > > We show our apology for the confusing description of the **response to W2.3**. We want to express that in our designed personalized strategy, each client only needs to upload the updated parameters of the encoders to the server for aggregation after local optimzation, which **is natural to the federated learning process without further fine-tuning or specifically trained on certain datasets to guarantee personalized prediction performance**.
> > > >
> > > > >C4. Difference with Fedrated Averaging.
> > > >
> > > > We agree with you that model parameters shared among clients are averaged in federated averaging algorithm. The main difference between Time-FFM and Federated Averaging lies in parameters of which parts are shared among clients. In our Time-FFM, we **merely aggregate the encoder parameters** at the server side by averaging mechanism, with **personalized prediction heads** for each client. While in Federated Averaging, all clients share **the whole model parameters**, i.e., the parameters of the encoders and prediction heads all need to be averaged, which fails to guarantee the personalized prediction performance (by comparing the **A.1 and A.4 in Table 5** of the manuscript).
> > > >
> > > > >C5. Further comparison with "(3) federated learning methods".
> > > >
> > > > We are sorry for the misunderstanding and for missing the comparison with the existing federated forecasting methods. In the **"relevant works"**, the proposed methods, such as FedGRU [3], FASTGNN [4], and CNFGNN [5], achieve inferior performance compared with the corresponding centralized versions, which perform more poorly than the SOTA centralized methods (i.e., the baselines in **TY2 and TY3**). Hence, we think our proposed Time-FFM would outperform these federated methods. We will carefully supplement the comparison analysis and experiments in the modified version. We hope such revision would be under your consideration.
> > > >
> > > > [5] Meng, Rambhatla, et al. Cross-node federated graph neural network for spatio-temporal data modeling. KDD. 2021.
> > > >
> > > > We will carefully incorporate your suggestions and the responses in the modified version. Considering the encouraging comments from other reviewers, we believe our research findings and technical novelty are worth sharing with the research community. We sincerely hope that the above discussion can release your concerns and a revision is still under your consideration.

---

> > > > > ### Comment · Reviewer_uvxC · 2024-08-13
> > > > > **Thanks for your rebuttal!**
> > > > >
> > > > > Dear Authors,
> > > > >
> > > > > Again, I appreciate the time and effort that you have put into the rebuttal.
> > > > >
> > > > > I really appreciate that you have tried to address all my comments in a limited time frame, and for this long and detailed discussion. I will raise my score, not because all of my core concerns have been addressed, but because I believe that the authors have made an honest effort at putting forth their side of the story. I think a broader discussion that involves the entire community is important to resolve my core concerns.
> > > > >
> > > > > Now onto my core concerns: there are 2 things that motivate this work (that I do not completely agree with):
> > > > > 1. the unavailability of permissively licensed public time series data that can be used to train foundation models for commercial use. Case in point is Nixtla's TimeGPT-1 / TimeGEN-1 which is public available time series forecasting foundation model trained on public data. Much like other foundation models, e.g., Meta's LLaMA, Google's Gemini, OpenAI's ChatGPT, TimeGPT too is trained on public data, of which some of it may be copyrighted or not permissively licensed. This has however not prevented these models from being productionized and made available publicly. To the best of my knowledge, the legality of this practice in sub judice. Also, for time series, many time series datasets in the Time Series Pile, LOTSA and Chronos' datasets are indeed permissively licensed.
> > > > > 2. I am still not aware of the mechanisms by which privacy leakage through time series data make occur. I skimmed through the papers that the authors cited, and I still did not find an answer to this question. Also, for any discussion of privacy, it is important to rigorously define the notion of privacy, e.g. differential privacy.
> > > > >
> > > > > I would strongly recommend that the authors add these discussions to the revised manuscript, along with the changes to related work and results.
> > > > >
> > > > > Good luck!

---

> > > > > > ### Author Response · Authors · 2024-08-13
> > > > > > **Thanks for Reviewer uvxC's response!**
> > > > > >
> > > > > > We show great gratitude for your willingness to increase the rating and support the recommendation for acceptance! We remain committed to contributing valuable insights to our field and are thankful for your constructive feedback throughout the review process.
> > > > > >
> > > > > > We make further clarifications on your core concerns as follows.
> > > > > >
> > > > > > 1. We admit that there are indeed a lot of permissively licensed public data that can be used for pre-training. However, the data scale of LOTSA is only 27B, while the case for Time Series Pile is merely 1B. We believe that this scale is still far from the commonly used pre-training datasets for LLMs. Furthermore, as the scaling law points out, due to data limitation, the size of time series foundation models is much smaller than that of LLMs, hence compromising the model's capabilities. Our preliminary research found that there are still many rich data that are hindered from being used due to privacy protection. Therefore, the method we proposed can incorporate more data while protecting data privacy.
> > > > > >
> > > > > > 2. On the one hand, the raw time series data may encompass personally-private information or reflect the commercial models, which the owners are reluctant to share the access to. Hence, we think the time series data are privacy-contained. On the other hand, we will incorporate the approach of differential privacy in the future works, to avoid the pretraining data detection from model parameters [1].
> > > > > >
> > > > > > [1] Shi W, Ajith A, Xia M, et al. Detecting pretraining data from large language models[J]. arXiv preprint arXiv:2310.16789, 2023.
> > > > > >
> > > > > > We will carefully follow your suggestions to add the discussions of your core concerns in the revision. We appreciate your insightful and valuable comments again, which is of great significance to clarify the motivation and novelty of our paper.

---

### Official Review · Reviewer_PDR4 · 2024-07-14

**Soundness:** 3
**Presentation:** 3
**Contribution:** 2
**Rating:** 5
**Confidence:** 4

**Summary:**

This paper introduces TIME-FFM, which is a Federated Foundation Model for Time Series Forecasting. TIME-FFM is comprised of (1) Modality Alignment which aligns time series patches with text tokens; (2) Prompt Adaption which learns the text prompts for an input time series; (3) LM backbone; (4) Prediction Head for domain specific output. Experiments are conducted on several benchmark datasets to demonstrate the effectiveness of the proposed method.

**Strengths:**

1. Federated learning for time series foundation model is an interesting and promising direction. This paper presents the first attempt in this direction.
2. Compared with SOTA federated methods (the TY1 group), the proposed method TIME-FFM could outperform the baselines.
3. The writing is clear and easy to follow.

**Weaknesses:**

1. The overall novelty is limited. Modality Alignment, Prompt Adaption and different prediction head has been explored by previous methods.
2. Compared with SOTA methods from TY2, the proposed TIME-FFM does not have a significant improvement.

**Questions:**

1. How do you obtain $E'$ from $E$?
2. Have you tried personalized Modality Alignment and Prompt Adaption for different domains?

---

> ### Author Rebuttal · Authors · 2024-08-05
>
> We express our gratitude to you for providing constructive feedback on our paper. We have addressed the specific concerns as detailed below.
>
> > W1: The overall novelty is limited. Modality Alignment, Prompt Adaption and different prediction head has been explored by previous methods.
>
> Thanks for your insightful concerns on our contributions. We aim to propose a language model-empowered federated foundation model for time series forecasting.
>
> - Given the differentiation of dimensionality and horizon, we introduce the modality alignment module encomposing the channel-independent and patching techniques, which may follow the track of GPT4TS, Time-LLM, MOIRAI, Moment, etc.
> - For bootstrapping the pre-trained GPT2 backbone for cross-domain time series reasoning, we propose to **adaptively construct prompts from how to understand patch tokens, rather than from rigid domain instructions** like Time-LLM and UniTime.
> - Due to cross-domain time series heterogeneity, we devise a personalized federated strategy (different from Federated Averaging which aims at learning a global prediction model), with **global encoder and personalized prediction heads**.
>
> In conclusion, we propose the first federated foundation model for time series forecasting, with adaptively generating domain-specific prompts and tackling time series heterogeneity for general-purpose learning and personzalied prediction.
>
> > W2: Compared with SOTA methods from TY2, the proposed TIME-FFM does not have a significant improvement.
>
> The methods in TY2 are centralized methods which intrinsically outperform federated methods with the same model structures. However, our proposed Time-FFM even outperforms the SOTA methods in TY2, which further indicates the effectiveness of the devised prompt adaption module and the personalized federated learning strategy.
>
> > Q1: How do you obtain $E$ from $E'$?
>
> This is accomplished through a simple linear projection. Specifcally, given $E \in \mathbb{R}^{V \times D}$, we learn a weight matrix $W \in \mathbb{R}^{V\times V'}$ to identify a small set of text prototypes in $E' \in \mathbb{R}^{V'\times D}$. We will revise the paper accordingly and include this technical detail in the revised version.
>
> > Q4: Have you tried personalized Modality Alignment and Prompt Adaption for different domains?
>
> Thanks you for such insightful suggestions. **A.6 Time-FFM-D** in Table 5 denotes the distributed version of Time-FFM, i.e., personalized modules of Modality Alignment, Prompt Adaption, and Prediction Head for each domain. As depicted in Table 5, the performance of **A.6** is inferior to that of **A.1 Time-FFM**, which further indicates that constructing a unified model across domains outperforms training dedicated prediction models for each domain.

---

> > ### Author Response · Authors · 2024-08-12
> > **Kindly Request for Reviewer's Feedback**
> >
> > Dear Reviewer PDR4,
> >
> > Since the End of author/reviewer discussions is coming in one day, may we know if our response addresses your main concerns? If so, we kindly ask for your reconsideration of the score. Should you have any further advice on the paper and/or our rebuttal, please let us know and we will be more than happy to engage in more discussion and paper improvements.
> >
> > Thank you so much for devoting time to improving our paper!

---

> > > ### Comment · Reviewer_PDR4 · 2024-08-13
> > >
> > > Thanks for your responses.
> > >
> > > My concerns are mostly addressed, and I would like to raise my scores.
> > > However, I still want to know more details about $E$ and $E'$, which I believe is a critical component for aligning time series and LLM.
> > > How do you identify a small set of text prototypes? Can you show some learned text prototypes? Which prototypes are important and which are not?

---

> > > > ### Author Response · Authors · 2024-08-14
> > > > **Thanks for Reviewer PDR4's response!**
> > > >
> > > > We show great gratitude for your willingness to increase the rating are thankful for your constructive feedback throughout the review process. We make further clarifications on your core concerns as follows.
> > > >
> > > > The linear layer **in the response to Q1** has the input dimension of $V$ the the output dimension of $V'$. Specifically, we first transpose $E \in \mathbb{R}^{V \times D}$ into $E^T \in \mathbb{R}^{D \times T}$ and then input $E^T$ into the linear layer to generate $E'^T \in \mathbb{R}^{D \times V'}$. Finally we transpose $E'^T$ into $E' \in \mathbb{R}^{V' \times D}$. That is to say, we do not change the dimension ($D$) of the word embeddings but the number ($V$ to $V'$ and **word embeddings to text prototypes**).
> > > > Hence, each text prototype is the weighted combination of all word embeddings.
> > > >
> > > > **Figure 3(a)** in the manuscript shows the the attention scores of between text prototypes (the X-axis) and the patch tokens (the Y-axis) in the all 8 attention heads. **Figure 3(b)-(d)** illustrate the distribution of top M (12 in our experiments) on different datasets. We have the key observation that **different datasets correspond to varying text prototypes**, which indicates the generated prompts are dataset-dependent.
> > > >
> > > > We will add analysis on the correlation between word embeddings and text prototypes in the modified version and also supplement the technical details of how to generate text prototypes. Considering the encouraging comments and increasing scores from other reviewers, we believe our research findings and technical novelty are worth sharing with the research community. We sincerely hope that the above discussion can release your concerns and a revision is still under your consideration.

---

> > > > > ### Comment · Reviewer_PDR4 · 2024-08-14
> > > > >
> > > > > Thank you for your clarification.

---

### Official Review · Reviewer_LqRs · 2024-07-18

**Soundness:** 3
**Presentation:** 3
**Contribution:** 3
**Rating:** 6
**Confidence:** 4

**Summary:**

The paper introduces TIME-FFM, a federated foundation model aimed at addressing the challenges of time series forecasting due to data scarcity and privacy concerns. The approach involves transforming time series data into text tokens, leveraging pretrained language models (LMs) for analysis, and using a personalized federated training strategy.

**Strengths:**

1.	Innovative Approach: The idea of transforming time series data into text tokens and leveraging LMs is creative and could potentially address the data scarcity issue effectively.
2.	Addressing Privacy Concerns: The use of federated learning helps in alleviating privacy concerns and encourages data sharing without compromising sensitive information.
3.	Personalized Training Strategy: The personalized federated training strategy, which combines global encoders with local prediction heads, is a thoughtful approach to handling data heterogeneity across domains.

**Weaknesses:**

1.	Algorithm Description: On page 6, the description of the algorithm seems to contain an error where Line 6 should actually be Line 5. Please carefully check the manuscript.
2.	Experimental Results Discrepancy: On page 6, the value of Baseline PatchTST on ETTm1 in Table 1 differs from the values reported in reference 25. You need to provide a detailed explanation of your simulation results and why these differences occur.
3.	Discussion on Future Directions: The article could benefit from a more in-depth discussion of the future directions and challenges for the proposed method.

[25] Xu Liu, Junfeng Hu, Yuan Li, Shizhe Diao, Yuxuan Liang, Bryan Hooi, and Roger Zimmermann. 369 Unitime: A language-empowered unified model for cross-domain time series forecasting. In 370 Proceedings of the ACM Web Conference 2024, 2024.

**Questions:**

Include a more detailed discussion on the potential future developments and challenges for TIME-FFM, highlighting areas for further research and potential improvements.
Provide a detailed explanation of the simulation results in Table 1, particularly regarding the discrepancies with reference 25. This will help in understanding the validity and reliability of your experimental results.
Correct the algorithm description to ensure that Line 6 reflects the appropriate content, as it currently appears to match what should be in Line 5.

**Limitations:**

Please refer to the described weakness above.

---

> ### Author Rebuttal · Authors · 2024-08-05
>
> We express our sincere thanks for the detailed and thoughtful review of our manuscript and for the encouraging appraisal of our work. We have addressed the specific concerns and respond to the constructive recommendations detailedly as follows.
>
> > **W1. Algorithm Description**: On page 6, the description of the algorithm seems to contain an error where Line 6 should actually be Line 5. Please carefully check the manuscript.
>
> Thanks for your careful readings. Line 1-5 in Algorithm describes the process of global execution and the server obtains the global model parameters by Line 5. Line 6 annotates that the following lines are meant "for local training". We are sorry for confusing you  and will polish the description in the revised version.
> > **W2. Experimental Results Discrepancy**: On page 6, the value of Baseline PatchTST on ETTm1 in Table 1 differs from the values reported in reference 25. You need to provide a detailed explanation of your simulation results and why these differences occur.
>
> As is described in the footnote of Page 5, we modify PatchTST into a "unified" version as per Reference [25]. Table 1 reports the averaged evaluation results over 4 prediction windows. The MSE and MAE values of PatchTST on ETTm1 dataset are "0.971, 0.629". In Reference [25], the averaged evaluation results of PatchTST in the type of "Models Trained Across Datasets" on ETTm1 are also "0.971, 0.629". Therefore, our simulation results agree with Reference [25].
> > **W3. Discussion on Future Directions**: The article could benefit from a more in-depth discussion of the future directions and challenges for the proposed method.
>
> We sincerely show great gratitude for your valuable suggestion.
>
> **For challenges**: We aim to propose a LM-empowered federated foundation model for time series forecasting, which is non-trivial technically, considering the following aspects.
>
> - **heterogeneous inputs**: Cross-domain time series data input into the foundation model are heterogeneous in terms of dimensions and historical readings, posing evident difficulty to modality alignment.
> - **Rigid instructions as prompts**: Prompts are adopted to bootstrap LMs for time series reasoning hinging on rigid domain-specific instructions, rather than the understanding of LMs, exhibiting poor robustness for unseen domains.
> - **Conflicts between generalization and personalization**: The ideal foundation model needs to learn the common temporal representations across domains and simultaneously enable the personalized prediction for domain-specific inputs.
>
> **For future directions**: Future directions include two aspects.
>
> - Thanks to the adopted "personalized prediction heads", future researches may focus on constructing a unified foundation model over different time series analysis tasks, such as classification, anomaly detection, and forecasting.
> - Further research should explore how to train a foundation model from scratch based on the existing and potential large-scale time series repositories.
>
> We will follow your suggestions and supplement more in-depth challenge analysis and future directions in the modified version.

---

> > ### Author Response · Authors · 2024-08-12
> > **Request for Reviewer's Feedback**
> >
> > Dear Reviewer LqRs,
> >
> > Since the end of author/reviewer discussions is coming soon (in one day), may we know if our response addresses your main concerns? If so, we kindly ask for your reconsideration of the score. Should you have any further advice on the paper and/or our rebuttal, please let us know and we will be more than happy to engage in more discussion and paper improvements.
> >
> > Thank you so much for devoting time to improving our paper!

---

> > > ### Author Response · Authors · 2024-08-13
> > > **Kindly Request for Reviewer's Acknowledge**
> > >
> > > Dear Reviewer LqRs,
> > >
> > > As the discussion phase is about to end and we were really trying our best to resolve your concerns, could you please acknowledge if your concerns are addressed? If so, please reconsider the rating; if not, we are very happy to resolve your further concerns. Thanks for your time.

---

### Author Rebuttal · Authors · 2024-08-05

We commerce by thanking the four reviewers for their thoughtful and constructive comments. We are really encouraged to see that the reviewers appreciate some positive aspects of our paper, such as technical quality (**Reviewer LqRs, PDR4, uvxC, and 7qPS**) and presentation skills (**Reviewer LqRs, PDR4, uvxC, and 7qPS**). Your expertise significantly helps us strengthen our manuscript. We are sorry for several unclear parts and weakness mentioned by reviewers and endeavor to respond to each comment. We sincerely hope that the responses can release the reviewers' concerns. We present a brief introduction of the responses as follows.

- In response to the feedback from **Reviewer PDR4, uvxC, and 7qPS**, we have clarified the novelty and contributions of our paper.
- In response to feedbacks from **Reviewer LqRs**, we have checked the details of the algorithm description and numerical results, and provided the in-depth analysis of the challenges and future directions.
- In response to feedbacks from **Reviewer PDR4**, we have clarified the experimental details and analysis of experimental results.
- In response to feedbacks from **Reviewer uvxC**, we have compared the performance with the suggested literature and analyzed the potential limmitations.
- In response to feedbacks from **Reviewer 7qPS**, we have detailedly analyzed the potential problems in practical application and supplemented the suggested performance comparison and analysis to validate the effectiveness of the proposed Time-FFM.

In the attachment PDF, we report the supplemented experimental results for **Reviewer uvxC and 7qPS**.

---

### Decision · Program_Chairs · 2024-09-25

**Decision:**

Accept (poster)

**Comment:**

The paper presents a federated foundation model for time series forecasting, based on tokenizing time series data for easy ingestion by LLMs and a federated training strategy for addressing privacy concerns.  The authors and reviewers engaged in significant discussion which clarified several points, especially around whether federated learning is in fact useful in this setting, and the reviewers are (reasonably) satisfied with the rationale given.

Multiple reviewers raised questions on novelty, pointing out that the paper is largely combining previous ideas in a new way.  Personally, I think that the vast majority of computer science in general is founded on the idea that prior pieces can be combined in new ways, and feel that the field of ML is no exception.  However, my personal feeling aside, the author response discussion and response show that the reviewers have been again satisfied with the clarifications -- and I very much encourage the authors to incorporate these points into the revision.

On the question of overall impact of the work, again reviewers raised some concerns that were well addressed during the rebuttal and response phase.  Personally, my view is that the direct impact of the work is likely to be limited -- it is not clear to me that the world has a burning need for a fully federated way to adapt LLMs to time series forecasting.  However, the work is technically strong, and even more importantly I think serves as an excellent example of how components like this may be combined, and how a resulting system can and should be fully evaluated.

As such, I believe this work is worthy of inclusion of NeurIPS, and agree with the reviewer consensus for accepting.  I very much encourage the authors to make full use of the extensive reviewer feedback to further strengthen the draft into its final form.